# Sulfate formation is dominated by manganese-catalyzed oxidation of $SO_2$ on aerosol surfaces during haze events

Weigang Wang [1,2,11], Mingyuan Liu [1,2,11], Tiantian Wang [3,11], Yu Song[3 ✉], Li Zhou[1], Junji Cao[4], Jingnan Hu[5], Guigang Tang[6], Zhe Chen[7], Zhijie Li[8], Zhenying Xu[3], Chao Peng[1], Chaofan Lian[1], Yan Chen[1], Yuepeng Pan [8], Yunhong Zhang[7], Yele Sun[8], Weijun Li [9], Tong Zhu[3], Hezhong Tian [10] & Maofa Ge[1,2 ✉]

The formation mechanism of aerosol sulfate during wintertime haze events in China is still largely unknown. As companions, $SO_2$ and transition metals are mainly emitted from coal combustion. Here, we argue that the transition metal-catalyzed oxidation of $SO_2$ on aerosol surfaces could be the dominant sulfate formation pathway and investigate this hypothesis by integrating chamber experiments, numerical simulations and in-field observations. Our analysis shows that the contribution of the manganese-catalyzed oxidation of $SO_2$ on aerosol surfaces is approximately one to two orders of magnitude larger than previously known routes, and contributes 69.2% ± 5.0% of the particulate sulfur production during haze events. This formation pathway could explain the missing source of sulfate and improve the understanding of atmospheric chemistry and climate change.

[1] State Key Laboratory for Structural Chemistry of Unstable and Stable Species, Beijing National Laboratory for Molecular Sciences (BNLMS), CAS Research/ Education Center for Excellence in Molecular Sciences, Institute of Chemistry, Chinese Academy of Sciences, Beijing, China. [2] University of Chinese Academy of Sciences, Beijing, China. [3] State Key Joint Laboratory of Environmental Simulation and Pollution Control, Department of Environmental Science, Peking University, Beijing, China. [4] Key Laboratory of Aerosol Chemistry and Physics, State Key Laboratory of Loess and Quaternary Geology, Institute of Earth Environment, Chinese Academy of Sciences, Xi'an, China. [5] Institute of Atmospheric Environment, Chinese Research Academy of Environmental Sciences, Beijing, China. [6] State Environmental Protection Key Laboratory of Quality Control in Environmental Monitoring, China National Environmental Monitoring Centre, Beijing, China. [7] The Institute of Chemical Physics, School of Chemistry and Chemical Engineering, Beijing Institute of Technology, Beijing, China. [8] State Key Laboratory of Atmospheric Boundary Layer Physics and Atmospheric Chemistry, Institute of Atmospheric Physics, Chinese Academy of Sciences, Beijing, China. [9] Department of Atmospheric Sciences, School of Earth Sciences, Zhejiang University, Hangzhou, China. [10] State Key Joint Laboratory of Environmental Simulation and Pollution Control, School of Environment, Beijing Normal University, Beijing, China. [11] These authors contributed equally: Weigang Wang, Mingyuan Liu, Tiantian Wang. ✉email: songyu@pku.edu.cn; gemaofa@iccas.ac.cn

As an important component of fine particulate matter ($PM_{2.5}$), sulfate exerts a significant influence on the Earth's climate system, air quality, and public health[1–3]. In recent years, severe haze episodes characterized by high sulfate concentrations have occurred frequently in northern China[4]. Hourly sulfate concentrations typically exceed 100 µg/m³ during wintertime haze events[5]. Traditionally, sulfate formation mechanisms primarily include the gas phase oxidation of $SO_2$ by OH radicals and the aqueous oxidation of S(IV) by $H_2O_2$, $O_3$, organic peroxides, and $O_2$ catalyzed by transition metal ions (TMIs), e.g., Fe(III) and Mn(II), in cloud/fog water droplets[6]. The large gap between observed and simulated sulfate concentrations during haze events, however, indicates the existence of unknown pathways for sulfate production[5,7].

Recently, several chemical mechanisms for sulfate production have been proposed and highlighted, mainly occurring in the aerosol liquid water. Cheng et al.[8] suggested that the S(IV) could be oxidized by $NO_2$ in aerosol water, and Liu et al.[9] proposed that the oxidation of S(IV) by $H_2O_2$ in aerosol water considering the ionic strength effect could be an important pathway. Besides, Song et al.[10] suggested that the hydroxymethanesulfonate (HMS) produced from the reaction of S(IV) with HCHO in aerosol water may explain the observed high levels of particulate sulfur.

Constrains exist in the above aqueous reaction routes, however. First, the aerosol water content (AWC) usually ranges from tens to hundreds of micrograms per cubic meter in heavy hazes, which is still multiple orders of magnitude less than the cloud water content ($0.05–3\ \mathrm{g\,m^{-3}}$)[6]. The space for aqueous reactions is too small to produce sulfate. Second, the aerosol water is often more acidic, with pH values ranging from ~4 to 5 in heavy hazes[11–13], which limits the dissolution of $SO_2$. Third, these reactions consume a large amount of photochemical oxidants when generating sulfate, but the oxidant amounts are not always sufficient in heavy hazes[14,15].

Xue et al.[16] and Wang et al.[17] suggested that reactions in cloud/fog water might contribute significantly to sulfate production. However, fog does not occur in most haze events[18,19]. Additionally, haze events often occur during stagnant weather conditions with stable stratification and weak turbulent diffusion[20,21]. Consequently, it is difficult to transport near surface precursors upward to high altitudes, and the sulfate generated in high-altitude clouds is also difficult to transport downward to the near surface. Therefore, in-cloud reactions could not account for sulfate formation during haze events. Indeed, numerical simulations have not indicated that the aqueous oxidation in cloud/fog water is an important pathway during haze events[5,7].

China is the country with the world's largest coal consumption, averaging approximately three billion tons of coal annually in recent years[22], and contributing nearly 80% of the national $SO_2$ emission[23]. As companion emitters, atmospheric heavy metals (including TMIs) from coal combustion are present in much higher concentrations in northern China than in developed countries[24–26]. Traditionally, S(IV) could be oxidized by TMIs in cloud/fog water[27,28].

In this work, we show that the manganese-catalyzed oxidation of $SO_2$ on aerosol surfaces dominates sulfate formation during haze events. The mechanism is identified via chamber experiments, and the sulfate formation rate of this mechanism is approximately one to two orders of magnitude larger than previously known routes. In-field observations show similar temporal variations, size distributions and internal mixing state of Mn and sulfate. Furthermore, chemical transport model simulations show that the manganese-catalyzed oxidation of $SO_2$ on aerosol surfaces dominates sulfate formation and contributes $92.5 ± 3.9\%$ of the sulfate ($69.2 ± 5.0\%$ of the particulate sulfur) production during haze events.

## Results and discussion

**Laboratory studies.** The $SO_2$ oxidation experiments were performed in a temperature and humidity-controlled chamber, which is described in detail in the "Methods" section (Supplementary Fig. 1)[29]. Size-selected particles (ammonium sulfate or sodium chloride) with different aerosol phase concentrations of $Mn^{2+}$ were added to the chamber along with various mixing ratios of $SO_2$ and $NH_3$. The rapid growth of both particle mass concentration and diameter were observed in the presence of $Mn^{2+}$ at 298 K (Fig. 1a), especially with sodium chloride seed particles. There was no apparent sulfate formation in the absence of any one of these factors, however, including $O_2$, aerosol phase $Mn^{2+}$, and aerosol liquid water (Supplementary Fig. 2). In addition, this reaction was found to be inhibited by more acidic seed particles (ammonium sulfate) and a lower concentration of $NH_3$, suggesting that pH could be another critical factor for sulfate formation. The calculated sulfate formation rate ranged from 14.1 to 24.5 $\mathrm{µg\,m^{-3}\,h^{-1}}$, while the mean diameter of seed particles could increase from ~60 to 82–105 nm within 50 min. This phenomenon could not be explained by the $Mn^{2+}$ aqueous catalysis pathway based on model calculation, even without considering the ionic strength inhibition effect (Fig. 1a). The sulfate formation rate via the aqueous route is approximately several orders of magnitude lower than the observed formation rate[6]. Harris et al.[30] also detected an unexpected rapid $SO_2$ oxidation rate in a dust leachate, which contained a mixture of TMI (Fe, Ti, Mn, and Cr) rather than the iron-only catalytic pathway. In our study, compared to the Mn-catalytic reaction, there was no noticeable sulfate formation observed through the Fe-catalytic pathway in the chamber experiment (Supplementary Fig. 3), and the reaction rate of the Fe–Mn-catalytic reaction did not show any synergism enhancement effect. This evidence indicated that the role of individual TMIs or their composite effect to sulfate formation need further reinvestigation under close to real atmosphere conditions. The heterogeneous reaction on the aerosol surface probably plays a vital role in this process.

To further evaluate the role of $SO_2$ and $Mn^{2+}$, chamber experiments were carried out with different concentrations of $Mn^{2+}$ and $SO_2$ under similar relative humidity (RH) and $NH_3$ conditions. As shown in Fig. 1b, the measured sulfate formation rates varied linearly with the concentrations of $Mn^{2+}$ and $SO_2$, suggesting the first-order reaction of both $SO_2$ and $Mn^{2+}$. A transmission electron microscopy energy-dispersive X-ray spectroscopy (TEM-EDX) was utilized to analyze the single-particle morphology and chemical composition at the micro-level (Fig. 1c). Most of the particles containing Mn generated a much larger size (a minimum of 100 nm) during chamber reactions (Supplementary Table 1) compared to the initial produced seed particle size (50 nm). N and S elements observed at the same location indicated the formation of ammonium sulfate. The particle morphology and elemental distributions of S, Mn, and N confirmed that ammonia sulfate formation was present Mn-catalytic reaction.

In the $Mn^{2+}$ catalytic aqueous phase reaction, Tursic et al.[31] found that substituting NaCl for $NaNO_3$ as the seed material could lead to a faster reaction under acidic (pH = 3) and dark conditions due to the formation of $Cl_2^-$ radical ions. Moreover, Harris et al.[32] found that $SO_2$ was rapidly oxidized on the NaOCl aerosol which produced HOCl in solution. To examine whether the simultaneous presence of $Mn^{2+}$, $O_2$, and $Cl^-$ in aerosol liquid water would facilitate HOCl formation, chamber experiments were also conducted under the $NaNO_3$ seed condition. As depicted in Supplementary Fig. 4, there was no clear reaction rate decrease in the $NaNO_3$ seed reaction.

The sulfate formation route via the Mn-catalytic reaction pathway that was found in the chamber experiments should be

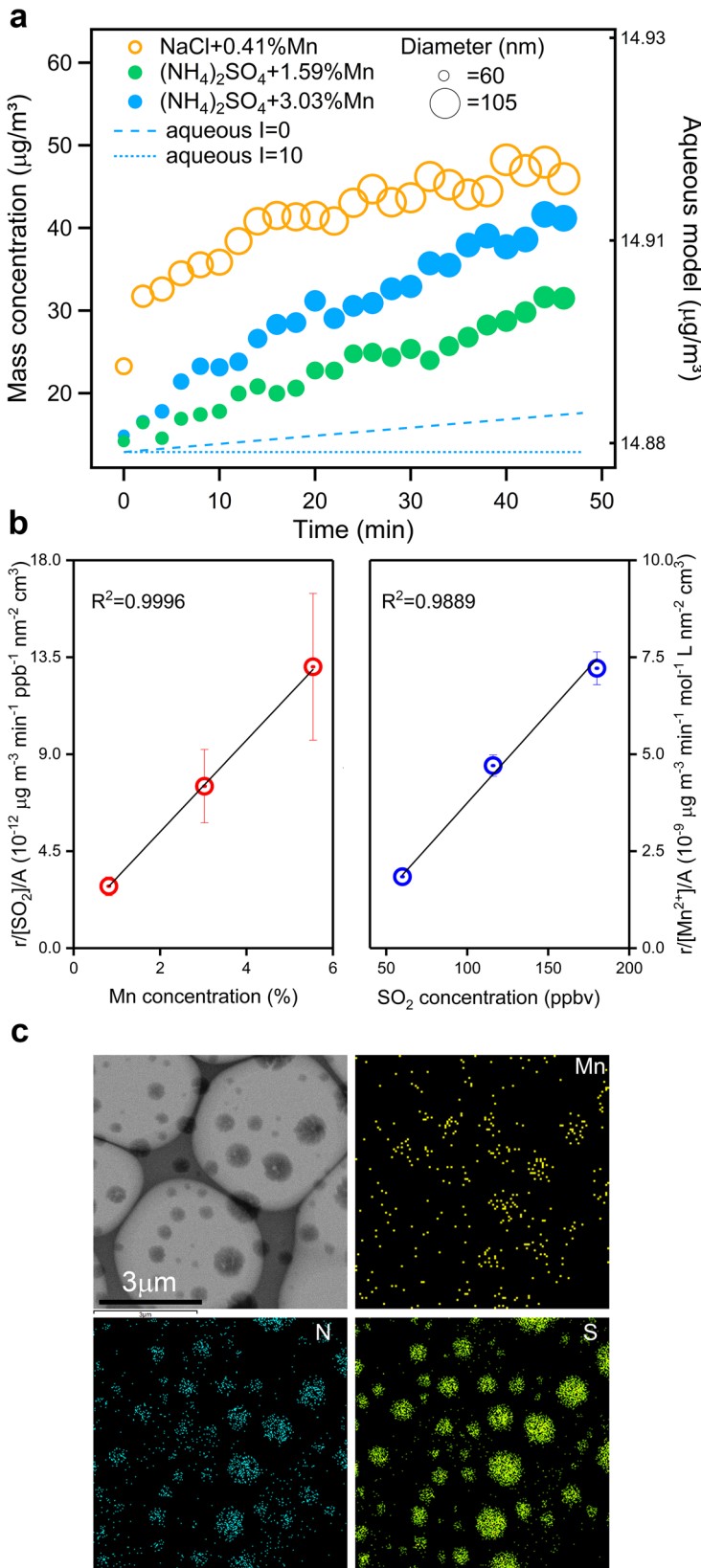

distinguished from the traditional aqueous routes. Therefore, in situ chemical composition measurements in single droplets using a confocal Raman microscope were performed to further evaluate this mechanism (Supplementary Fig. 5). Supplementary Fig. 7 presents the Raman spectrum changes of the ammonium

chloride and manganese chloride mixture in a droplet exposing to $SO_2$ and zero air, in which a noticeable increase in the peak intensity of sulfate was observed with increasing exposure time. As depicted in Fig. 2, parallel experiments were performed on droplets with diameters ranging from 5 to 50 μm. The sulfate peak

**Fig. 1 Sulfate formation via Mn-catalytic reaction in chamber experiments. a** Mass concentration and mean diameter growth of NaCl and $(NH_4)_2SO_4$ seed particles via the Mn-catalytic reaction at 298 K. In the NaCl experiment (open orange circles), the initial $Mn^{2+}$ concentration was 49.7 ng m$^{-3}$, the $SO_2$ mixing ratio was 116 ppbv, and the $NH_3$ mixing ratio was 84 ppbv under 80% relative humidity (RH). For the $(NH_4)_2SO_4$ experiment, the initial Mn concentration was 450.4 ng m$^{-3}$ (solid blue circles) or 225.4 ng m$^{-3}$ (solid green circles), and the $SO_2$ and $NH_3$ mixing ratios were the same as those of the NaCl experiment under 89% RH. The dotted and dashed blue lines depict the particle mass concentrations calculated by the Mn-catalytic aqueous phase reaction with and without considering the ionic strength effect. The initial conditions were the same as those of the solid blue circles. **b** Dependence of sulfate formation rate ($r$) on Mn (excluding impacts of $SO_2$ mixing ratios and surface area concentration ($A$), open red circles) and $SO_2$ mixing ratios (excluding impacts of Mn concentration and surface area concentration, open blue circles). Error bars represent standard deviation. **c** TEM and mapping images of particles collected from the chamber experiments.

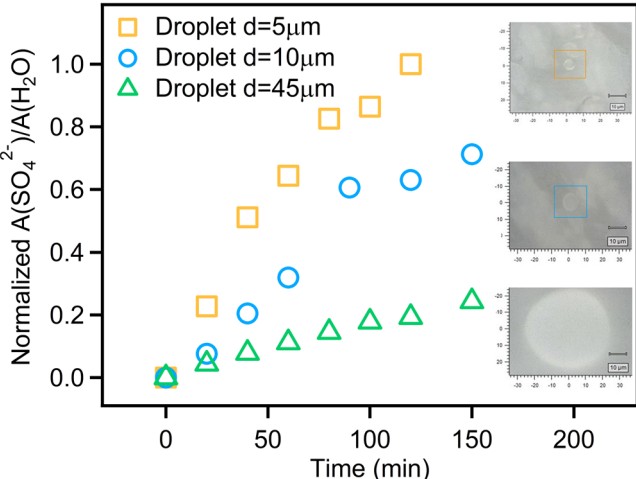

**Fig. 2 Raman peak area ratio of sulfate to water as a function of reaction time, as well as microscope images of droplets.** Relationships between sulfate area ratio and droplet diameter, as well as ×50 optical microscope images of droplets. $A$ represents Raman peak area, and $d$ represents diameter. All of these experiments were performed at 298 K, 85% relative humidity (RH), and 327 ppbv $SO_2$.

area ratio $A(SO_4^{2-})/A(H_2O)$ with standard solution calibration, representing the sulfate content of these three sets of experiments, exhibited a distinct increasing trend. The sulfate production rate increased with decreasing droplet diameter, revealing that the reaction rates were correlated with the surface-area-to-volume ratio, consistent with the heterogeneous reaction process[33]. These results verified the important role of the aerosol surface in sulfate formation during the chamber experiments.

The chamber experiments were carried out in a wide range of temperatures, from 298 to 278 K to study the temperature dependence of sulfate production rates. As illustrated in Fig. 3a, in high RH (>84%) conditions, the sulfate production rate decreased with decreasing temperature. This phenomenon is consistent with an exothermic reaction. The sulfate formation reaction would tend to acidify the aerosols, and $NH_3$ would inhibit the hydrolysis of ammonium, thus influencing the $H^+$ concentration in the aerosol phase. Consequently, the sulfate production rate was negatively correlated with $H^+$ concentration which was fitted in the form $f(H^+)$, as shown in Fig. 3a and Eq. (5) ("Method" section). The pH was calculated using the Extended Aerosol Inorganics Model IV[34]. We further investigated the influences of ionic strength on sulfate formation at 298 K (Fig. 3b). Unlike the restraints of TMI-catalyzed aqueous phase reactions under high ionic strength condition[9,35], there was no apparent decrease with increasing ionic strength, contrast to the ion–ion aqueous phase reaction mechanism. Meanwhile, under low-temperature conditions (278 and 283 K), accelerated sulfate formation rates were observed when the RH was lower

than 84% and 82%, respectively. The enhancement factors were ~10.0 ± 1.5 (278 K) and 10.6 ± 0.11 (283 K) when the aerosol ionic strength exceeded 14.2 and 15.3 mol l$^{-1}$, respectively. These thresholds were positively correlated with temperature, and the sulfate formation enhancement factors were similar (Fig. 3b). In addition, the experimental investigation of the $NO_2$ and $O_3$ pathways under low-temperature and high-ionic strength conditions revealed no apparent formation of sulfate (Supplementary Fig. 8). The sulfate formation rate was no more than the room temperature aqueous phase reaction rate, and the ionic strength effect on the $NO_2$ pathway was almost negligible.

**Mechanism of Mn-catalyzed reaction on aerosol surfaces.** Based on the above phenomena and the determined heterogeneous reaction mechanism, the following mechanism of the Mn-catalytic redox reaction on aerosol surfaces is proposed:

$$SO_2 + O_2 + Mn(OH)_x^{(3-x)} \rightarrow SO_5^{\cdot-} + Mn^{2+}, x = 1, 2 \quad (1)$$

$$SO_5^{\cdot-} + Mn^{2+} + H^+ \rightarrow Mn(OH)_x^{(3-x)} + HSO_5^-, \quad (2)$$

$$NH_3 + HSO_5^- + SO_2 + H_2O \rightarrow NH_4^+ + SO_4^{2-}. \quad (3)$$

Schematically, $SO_2$ is firstly absorbed at the surface layer of droplets containing dissolved $Mn^{2+}$ and seed species (ammonium sulfate or sodium chloride in this work). Then, the Mn-catalytic reaction rapidly occurred at the surface layer, and the formed sulfate is finally dispersed throughout the liquid phase (Fig. 4). In these reactions, Mn(III) is the intermediate product that could oxidize S(IV). Due to the high value of the hydrolysis equilibrium constant of $Mn^{3+}$, Mn(III) mainly exists in the forms of $Mn(OH)_2^-$ and $Mn(OH)^{2-}$, and their concentrations are correlated with aerosol phase acidity and Mn(II) concentration. Under high ionic strength conditions, high electrolyte concentrations may accelerate the reaction rate in ion–neutral molecular reactions by forming an association or ion pair as a new activation center[36]. The energy barrier of these intermediate products might be influenced by temperature changes, leading to a temperature-related enhancement effect of the ionic strength threshold.

The Mn-catalyzed aqueous phase oxidation of S(IV) has been extensively investigated and the critical oxidation process is the reaction between Mn(III) and hydrogen sulfite or its complex with Mn(II). Then $SO_3^-$ radicals can react with dissolved $O_2$ to generate $SO_5^-$ radicals, which can oxidize Mn(II) to regenerate Mn(III), and the rate of the reaction is independent of the oxygen concentration[6]. The bulk manganese catalytic reaction is the ion–ion reaction, in which rate constant decreases dramatically with increasing ionic strength, influenced by the primary kinetic salt effect[36]. The enhanced reaction rate for Mn-catalytic reaction on the aerosol surface compared to aqueous reaction is mainly attributed to several differences: the difference between the ion–neutral molecule and ion–ion reactions, which perform differently under high ionic strength; the different reaction space;

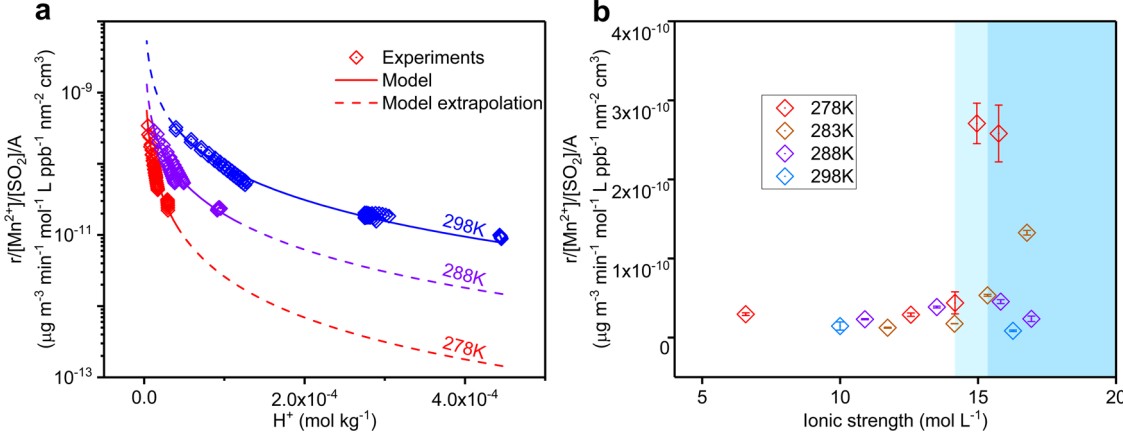

**Fig. 3 Temperature and ionic strength impacts on Mn-catalytic reaction. a** Temperature effect on the reaction, sulfate formation rate ($r$) excluding impacts of $Mn^{2+}$ concentration, $SO_2$ mixing ratio and particle surface area concentration ($A$) (see Eqs. (4) and (5)) is shown on the $y$-axis and the concentration of $H^+$ is shown on the $x$-axis. The blue, violet, and red symbols indicate the experiments conducted at 298, 288, and 278 K, respectively. All of the experiments at 278 K were conducted in >84% relative humidity (RH) conditions, while the experiments at 288 and 298 K were conducted in >80% RH. **b** Ionic strength effect on the reaction; the light blue shaded area indicates RH ranging from 82 to 84%; the blue shaded area indicates RH < 82%. Error bars represent standard deviation.

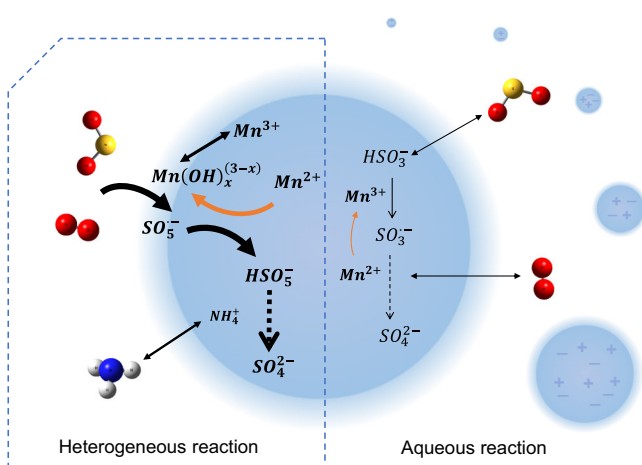

**Fig. 4 Schematic of Mn-catalytic oxidation of $SO_2$ on the aerosol surface and in the aqueous phase.** Red, blue, and white spheres represent oxygen, nitrogen, and hydrogen atoms, respectively.

and the surface of the aerosol and bulk solution. The AWC is scarce compared with cloud water content, and the low pH values limit the solubility of $SO_2$. The surface-area-to-volume ratio increases with the decreasing of the aerosol diameter, making the chemistry on the surface more important than in the bulk phase. Furthermore, Yan et al.[37] and Zhang et al.[38] found that reaction occurring in microdroplets obtained a higher rate than the same reaction in bulk solution. Hung et al.[39] proposed sulfate formation in the reaction of sulfurous acid microdroplet and oxygen, without additional oxidants, and the finding of spontaneously $H_2O_2$ production in the atomized bulk water microdroplets by Lee et al.[40] may shed light on this phenomenon. These findings demonstrated that there may be a massive gap in the reaction rate between air–liquid interface and bulk solution, and the mechanism of air–liquid interface rate enhancement effect may differ greatly in a distinct reaction system.

The experimental results revealed that the sulfate formation rate was correlated with aerosol acidity, $Mn^{2+}$ concentration, and $SO_2$ concentration. Therefore, sulfate formation in units of $\mu g\ m^{-3}\ min^{-1}$ of the Mn-catalytic reaction on the aerosol surface

could be denoted as:

$$\frac{d\left[SO_4^{2-}\right]}{dt} = k \times f(H^+) \times f(T) \times f(I) \times \left[Mn^{2+}\right] \times \left[SO_2(g)\right] \times A \quad (4)$$

where $k$ is the rate constant, $f(H^+)$ is the function of $H^+$, $f(T)$ is the function of temperature, $f(I)$ is the enhancement factor of ionic strength, $[Mn^{2+}]$ is the $Mn^{2+}$ concentration, $[SO_2(g)]$ is the $SO_2$ concentration in the gas phase, and $A$ represents the particle surface area concentration (details are provided in the "Methods" section).

**In-field observations**. Six long-lasting haze episodes occurred over the North China Plain (NCP) during January 2015 and December 2016 ("Methods" section). Approximately 90% of the days during these haze episodes were accompanied by temperature inversions, which were primarily surface-based with inversion layers extending from the surface to heights of 500–1000 m (Supplementary Fig. 9). Temperature inversions lower the vertical transfer of momentum, heat, and moisture, resulting in high humidity and weak winds[41,42]. The average RH during these haze episodes was $65 \pm 23\%$ and the wind speeds were generally <3 m s$^{-1}$. The stagnant conditions during haze episodes not only concentrate primary pollutants near the surface, but also favor the heterogeneous formation of secondary aerosols[43].

Several in-field observations were conducted to build a relationship between the proposed pathway and haze episodes. As shown in Supplementary Fig. 10, we analyzed the correlation between Mn and sulfate concentration in the Northern China Plain (Baoding, Tianjin, Beijing, Tangshan, Dezhou, and Xinxiang) from Dec 1st to 22nd, 2016. The concentration of Mn was within the range of 21.3–91.9, 14.1–105.6, 14.5–107.6, 20.4–327.0, 31.3–92.7, and 29.1–123.6 ng m$^{-3}$, respectively. In all sampling sites, mass concentration of Mn showed a similar variation trend as that of sulfate, with the exception of a small part of the sulfate rise and drop. Furthermore, we conducted water-soluble Mn concentration in the NCP during winter. Compared with total Mn, water-soluble Mn precisely tracked all of the sulfate variation trends in the observation period (Fig. 5). Since these results were consistent with our prior experiments, they provided solid evidence that dissolved Mn played a key role in sulfate formation. The size distribution of Mn and sulfate content sampled in wintertime Beijing in 2013, 2014, and

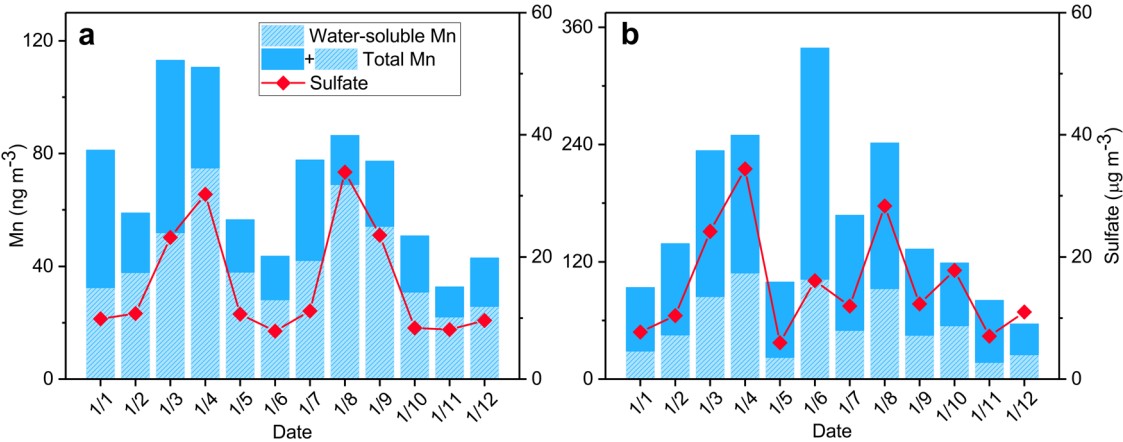

**Fig. 5 Temporal variation of water-soluble Mn, total Mn, and sulfate concentration in $PM_{2.5}$ in the North China Plain (Baoding and Tianjin) from January 1–12, 2015. a** Refers to Baoding, and **b** refers to Tianjin. Metal ions and sulfate concentration were measured by inductively coupled plasma mass spectrometry and ion chromatography, respectively.

2016 showed a similar variation trend in accumulation mode particles during different polluted periods (Supplementary Fig. 11), implying that the formation process of sulfate could be related closely to Mn. Also, we found the presence of Mn in Fe-rich particle of the mixture of Fe-rich and S-rich particle collected in wintertime Beijing and measured by TEM-EDX (Supplementary Fig. 12), although the detection limit of EDX restricted the detection of trace Mn in the S-rich part. Fe detected in the S-rich part could also indicate the existence of dissolved metal in sulfate because the solubility of Mn was greater than Fe in AWC. The TEM and element maps of samples collected in NCP (Shijiazhuang) showed that Fe and Mn were surrounded by S (Supplementary Fig. 13), indicating sulfate formation occurs with the particle containing Fe and Mn. Moreover, Mn appeared to be dispersed in the particle, which proved that Mn and sulfate were internally mixed, indicating that the formation of sulfate could happened with Mn. Using ambient $PM_{2.5}$ chemical component data and positive matrix factorization method[44], source apportionment results showed that more than 80% Mn were emitted from coal combustion during haze events (Supplementary Fig. 14).

**Chemical transport model simulations**. The three haze episodes in December 2016 were selected for further detailed analysis. The observed daily mean $SO_4^{2-}$ concentration was $31 \pm 17 \, \mu g/m^3$. The Weather Research and Forecasting model coupled with Chemistry (WRF-Chem) was used to investigate the sulfate formation mechanism ("Methods" section). Similar to the results of previous studies[5,7], the traditional pathways (gas phase and in-cloud oxidation) only explained 10% of the sulfate in the regional model (Experiment I) (Fig. 6a, d). We discovered that the Mn-catalytic reaction on aerosol surfaces (Experiment II) significantly improved sulfate simulation on the regional scale (Fig. 6b). The gap of $SO_4^{2-}$ values between Experiment I and observations was reduce considerably in Experiment II (NMB = −2.8%) (Fig. 6d). Similar results were also found for the three haze episodes in January 2015 (Supplementary Fig. 15). The simulated concentrations of other inorganic constituents in $PM_{2.5}$ ($NO_3^-$, $NH_4^+$, Fe, and Mn; Supplementary Figs. 16 and 17) and gaseous species ($SO_2$, $O_3$, and $NO_2$; Supplementary Fig. 18) were also comparable with observations ("Methods" section).

The Mn-catalytic reaction was supported by the high concentrations of $SO_2$, Mn, and aerosol surface area. The modeled aerosol surface area concentration was $>10 \, cm^2 \, m^{-3}$ over the NCP, providing ample spaces for the surface reaction (Supplementary

Fig. 19a). In addition, the high concentrations of $SO_2$ ($81 \pm 61 \, \mu g/m^3$) and Mn ($90 \pm 49 \, ng/m^3$) provided abundant reactants and catalysts for the reaction. It should be noted that only the dissolved Mn(II) in the liquid phase of deliquesced aerosols played a catalytic role. The high RH during the haze episodes was beneficial to the deliquescence of aerosols, and the mean AWC reached as much as $50 \, \mu g/m^3$ over the NCP (Supplementary Fig. 19b). Moreover, the aerosol pH over the NCP ranged from 4.0 to 4.6 (Supplementary Fig. 19c), which was comparable with the values found in previous studies[11–13]. The acidic deliquesced aerosols solubilized the Mn so it could then act as a catalyst[45,46].

The recently highlighted sulfate formation routes were also evaluated using the WRF-Chem model in Experiment III, involving the aerosol liquid phase reaction of S(IV) with $O_3$, $H_2O_2$, $NO_2$, and $O_2$ catalyzed by TMIs. The simulated sulfate concentration in Experiment III is almost equivalent to Experiment II (Fig. 6c, d). Sulfate formation during haze events is dominated by the Mn-catalytic oxidation on aerosol surfaces, accounting for $92.5 \pm 3.9\%$ of the sulfate production (Fig. 6e). Although the reaction rate of S(IV) with $H_2O_2$ in aerosol water at high ionic strength could be much higher[9], it only contributed $4.0 \pm 3.7\%$ to sulfate production. The simulated regional mean $H_2O_2$ decreased from 0.7 ppb in Experiment II to 0.1 ppb in Experiment III (an 85% decrease), indicating that the limited production rate of $H_2O_2$ could not support the fast reaction of $H_2O_2$ with S(IV). Gas phase oxidation of $SO_2$ by OH radicals is the third most important oxidation pathway, accounting for $3.2 \pm 0.6\%$ of sulfate production. The contributions of aerosol liquid phase S(IV) oxidation by $O_3$ ($0.2 \pm 0.1\%$) and $NO_2$ ($0.1 \pm 0.1\%$) were minor. There were two main reasons for this. First, the reaction rate of S(IV) with $O_3$ and $NO_2$ was low under the aerosol pH of 4.0–4.6. Second, even if the rate of this reaction could support the rapid sulfate yield, it would consume an equivalent amount of $O_3$ and $NO_2$. In this study, the modeled $O_3$ and $NO_2$ concentrations matched the observations well (Supplementary Fig. 18). Thus, this would not be an effective sulfate formation pathway. The TMIs-catalyzed $SO_2$ oxidation reactions in aerosol water also contribute little to sulfate. Three schemes of Mn(II) catalyzed-, Fe(III) catalyzed-, and the synergistic Fe(III)–Mn(II)-catalyzed oxidation were tested, and their contributions were $0.09 \pm 0.05\%$, $0.02 \pm 0.02\%$, and $0.05 \pm 0.08\%$, respectively. The low concentration of $Fe^{3+}$ in the moderately acidic aerosol water is not conducive to the catalytic reactions of iron. In addition, the rates of the Mn(II)-catalyzed and the synergistic Fe(III)–Mn(II)-catalyzed oxidation in liquid phase decreased exponentially

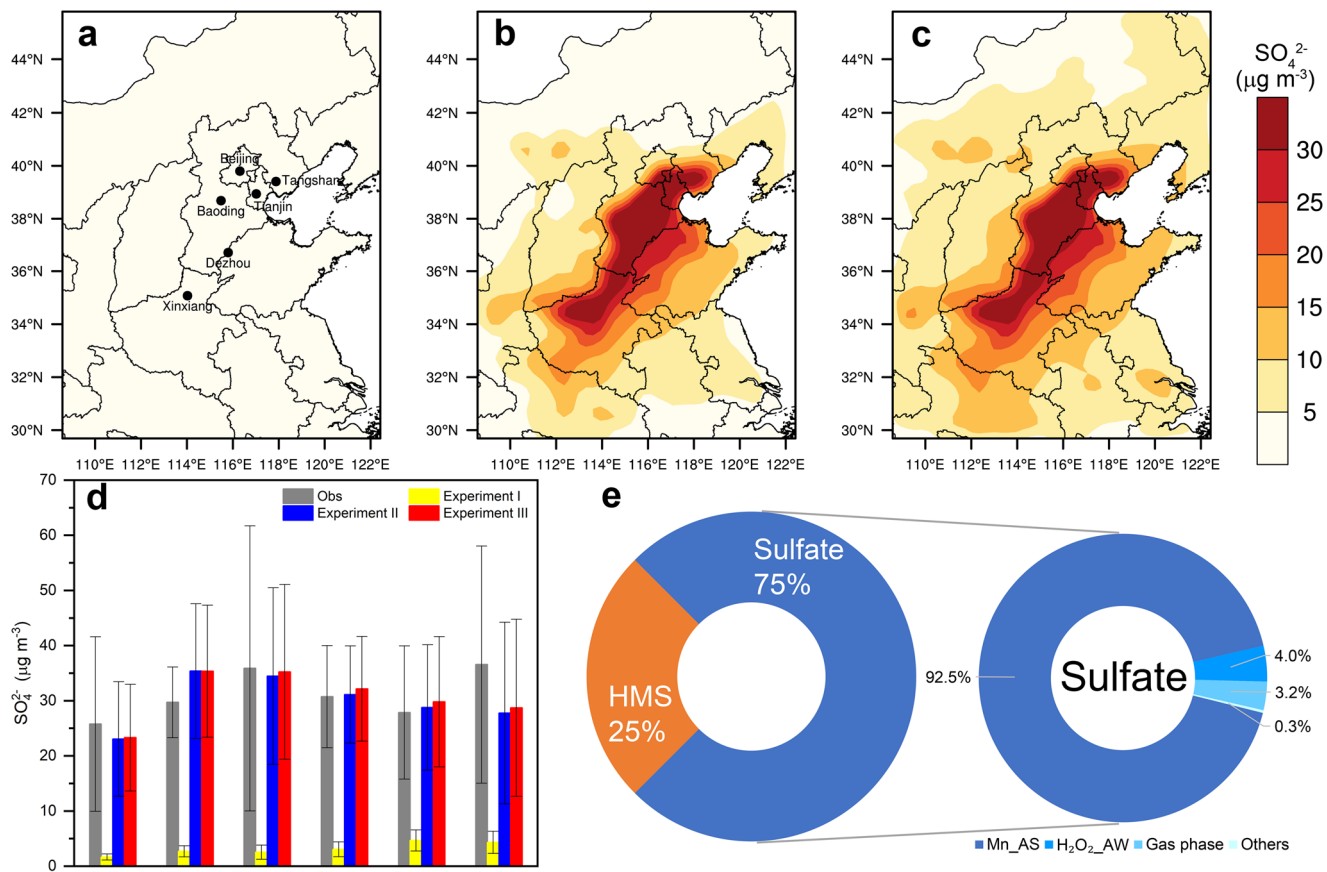

**Fig. 6 Observed and WRF-Chem simulated sulfate concentrations and contributions of different sulfate formation pathways. a** Spatial distributions of WRF-Chem simulated sulfate concentrations in Experiment I during the three haze episodes in December 2016. **b** Same as **a** but for Experiment II. **c** Same as **a** but for Experiment III. **d** Comparison of the observed and WRF-Chem simulated sulfate concentrations during the three haze episodes in December 2016 at six sites, the locations of which are indicated by black dots in **a**. Error bars represent standard deviation. **e** Composition of particulate sulfur and contributions of different sulfate formation pathways to sulfate production during the six haze episodes in January 2015 and December 2016, including Mn-catalyzed oxidation of $SO_2$ on aerosol surfaces (Mn_AS), reaction of S(IV) with $H_2O_2$ in aerosol water($H_2O_2$_AW), gas phase oxidation (Gas phase), and other sulfate formation pathways (Others, including in-cloud oxidation and reactions of S(IV) with $O_3$, $NO_2$, and $O_2$ catalyzed by TMIs in aerosol water). The contributions of each sulfate formation route were spatially averaged over the region with sulfate concentration >20 μg/m³. Maps were created by the authors using NCAR Command Language software Version 6.6.2[71].

with increasing ionic strength[6,47] and were suppressed by high ionic strength in aerosol water (30–50 mol l⁻¹; Supplementary Fig. 19d).

Additionally, Song et al.[10] suggested that HMS was usually misidentified as sulfate in previously observations and could contribute to the observed high particulate sulfur. In Experiment III, this mechanism was also considered as a sink of $SO_2$. Results showed that HMS from this pathway contributed 25.0 ± 7.8% (Fig. 6e) to the simulated particulate sulfur, which is slightly higher than the field observations during haze episodes in winter 2014 (17 ± 7%)[10]. If the particulate sulfur is composed of sulfate and HMS, sulfate might account for about 75% of the particulate sulfur concentration, with 92.5 ± 3.9% come from the Mn-catalytic oxidation of $SO_2$ on aerosol surfaces. Thus, the Mn-catalytic heterogeneous reaction might contribute 69.2 ±5.0% to the particulate sulfur. Consistently, stable sulfur isotope results supplied solid field observation evidence that TMI-catalytic reaction can be the dominant pathway for sulfate formation. TMIs contributed 49 ±10% to sulfate formation in winter 2015 in Nanjing[48].

**Atmospheric implications**. The rapid production mechanism of sulfate aerosols during winter haze episodes over the NCP has

been an unsolved problem for a long time. Significantly, most of the previous studies had focused on mechanisms in the aqueous phase[6,8–10,35,49]. Nevertheless, aqueous phase oxidation processes, such as oxidization by $NO_2$, $O_3$, $H_2O_2$, and so on, are not only limited by the availability of oxidants in the atmosphere and the solubility of $SO_2$, but these oxidants would also be consumed at the same order of magnitude as the formation of sulfate. The Mn-catalytic redox reaction, however, could continue to form sulfate while consuming only oxygen and $SO_2$, meaning that this reaction would occur throughout the sulfate formation process, from clean air to heavy haze events. The comparison between all known major sulfate formation routes and the discovered Mn-catalytic redox process illustrated that the Mn-catalytic heterogeneous reaction on aerosol surfaces dominates sulfate formation, and this mechanism significantly improves the model simulation on the regional scale. Identification of the Mn catalysis sulfate formation pathway provides perspectives on aerosol formation in the atmosphere, emphasizing the necessity for a comprehensive understanding of the reactions on aerosol surface, and perhaps oxidation on droplet surfaces in the cloud as well. This mindset is widely applicable across various ambient conditions, and the awareness of this mechanism will be useful in development of pollution control strategies in China and other countries.

## Methods

**Chamber system.** Chamber experiments were used to study the effects of different parameters on sulfate formation, including temperature, RH, gaseous precursor concentrations, and seed particles. Our chamber system (Supplementary Fig. 1) has three main parts: the injection, reaction, and detection parts. The injection part consists of a particle generator and a gas generator. Our previous work describes the chamber in more detail[29,50]. To produce suitable seed particles, different inorganic salt solutions were prepared and added to the atomizer (MSP 1500; MSP), and atomized using a high inlet pressure of air (737 series pure zero air generator; AADCO Instruments) or nitrogen (liquid nitrogen; Beijing Tianxinde Technology Development). Then, polydisperse seed particles were dehydrated through two-stage homemade diffusion dryers stuffed with silica gel (RH < 10%) and entered the electrostatic classifier (EC; model 3082; TSI)/differential mobility analyzer (DMA; model 3085; TSI). By applying a fixed voltage for a given sheath flow rate, monodisperse aerosols were generated from the EC-DMA based on their electrical mobility. The charged particles were neutralized by another X-ray source. Then, the particles passed through the electrostatic precipitator, which removed any remaining charged particles. Only monodisperse neutral particles were produced, thus avoiding significant electrical wall loss of charged particles. Then, the seed particles flowed through the humidifier (MH-110-24P-4; Perma Pure) under a high RH (>80%) to ensure that they remained in a deliquescence state. Conductive silicone tubing (TSI) was used for particle transmission, to minimize particle loss in the tubing wall. Sulfur dioxide ($SO_2/N_2$ 30 ppmv; Beijing Huayuan), ammonia ($NH_3/N_2$ 90 ppmv; Beijing Huayuan), and nitrogen dioxide ($NO_2/N_2$ 54.5 ppmv; Beijing Huayuan) were injected into the chamber through a mass flow controller (CS200; Sevenstar) or homemade transfer line. The reactions were conducted in a 200-L Teflon chamber, which was placed in a thermally isolated space to keep the temperature stable within ±0.2 K in each experiment. All experiments were performed with a RH greater than 80%, which is the deliquescent point of $(NH_4)_2SO_4$, except the experiments used to evaluate the role of AWC (Supplementary Table 1). Before each set of experiments, the chamber was thoroughly flushed with zero air until a number concentration lower than three particles $cm^{-3}$ was attained and the concentration of gaseous precursors was below the detection limit (<1 ppbv).

The detector is equipped with a series of instruments that sample from the center of the chamber to measure reactant and product variation during the reactions. A scanning mobility particle sizer (SMPS) system was used to determine the particle size distribution and mass concentration. The SMPS consists of an EC (model 3080; TSI), DMA (model 3081; TSI), and condensation particle counter (model 3776; TSI). A time of flight aerosol chemical speciation monitor (ACSM; Aerodyne Research) was used to measure the aerosol chemical composition; ammonium sulfate formation was confirmed by the ACSM and TEM-EDX. A $SO_2$-$H_2S$ analyzer (model 450i; Thermo Scientific), NOx analyzer (model T200UP; Teledyne), and $O_3$ analyzer (model T400; Teledyne) were used to measure the $SO_2$, $NO_2$, and $O_3$ concentrations, respectively. The $NH_3$ concentration was measured by proton transfer reaction quadrupole mass spectrometry (PTR-QMS; Ionicon Analytik) in $O_2^+$ mode.

**Micro-Raman system.** For micro-Raman measurements, a single droplet was used to evaluate the sulfate formation mechanism of the Mn-catalytic pathway. The instrument used has been described in detail elsewhere[51,52]. Briefly, a micro-Raman spectrometer consists of a sample chamber coupled with an RH controller, a gaseous precursor generator, an optical microscope (DMLM; Leica) for observing droplet morphology, and a confocal Raman spectrometer (inVia; Renishaw) with a 514.5-nm argon-ion laser (model LS-514; Laser Physics) as the excitation source, with a power of 30 mW for measuring Raman spectra. In each set of measurements, the spectrum was first calibrated using the 520 $cm^{-1}$ silicon band as the reference. Then, 3 mol/l ammonium sulfate solution mixed with 0.03 M manganese chloride was atomized on polytetrafluoroethylene (PTFE) substrate to form droplets, and a low flow (1 l/min) of humidified zero air or $N_2$ was injected to maintain a stable RH (85 ± 1%) in the sample chamber. Droplets were sitting on the PTFE substrate and had a nearly spherical shape and a contact angle of 112.59° (Supplementary Fig. 6) measured by an optical tensiometer (Theta Flex, Biolin Scientific). Using the 50× objective lens of the optical microscope, a laser beam was focused on the selected droplet. The dispersed Raman signals were detected through a strong-Rayleigh-scattering-removal notch filter and an 1800 g/mm grating, and recorded by a charge-coupled device.

The Raman spectrum of sulfate ion has four bands, which are all discernible in Supplementary Fig. 7c. Since the peak intensity of the symmetric stretching vibration was much stronger, we used the intensity of the 978 $cm^{-1}$ Raman band to evaluate the amount of sulfate in the selected droplets. The broad peak at around 3430 $cm^{-1}$ was identified as the OH stretching vibration of water. As illustrated in Supplementary Fig. 7a, when $N_2$ was used as the background gas (300 ppbv $SO_2$ in $N_2$), no sulfate formation was observed over 180 min. However, when zero air was used instead of $N_2$, the sulfate band of the droplet was detected at 978 $cm^{-1}$, and the peak intensity increased with time (Supplementary Fig. 7b). This confirmed that gas phase oxygen is indispensable in this reaction, consistent with the chamber experiments.

Raman spectroscopy was not sufficiently accurate for quantitative analysis, so a standard internal method was used with known species[53,54]. In this work, the ratio of the integrated peak areas of the symmetric stretching vibrations of sulfate and

OH gave the relative sulfate content. To eliminate the size effect, we measured the area ratio between sulfate and the OH stretching vibration in different-sized droplets (5, 10, and 45 μm), with the same sulfate concentration, to obtain the normalized peak area ratio. Throughout the reaction process, the sulfate content on the surface and in the center of the droplet remained the same, demonstrating that sulfate ions diffused quickly and reached equilibrium in the droplet once formed. Therefore, measurements at the center of the droplet represented the concentration for the entire droplet in our experiments.

**Equation of Mn-catalytic reaction rate.**

$$f(H^+) = -1/\left(1 + a[H^+] + b[H^+]^2\right), \tag{5}$$

$$f(T) = e^{-\frac{E}{R}\left(\frac{1}{T} - \frac{1}{T_0}\right)}, \tag{6}$$

$$f(I) = \begin{cases} 1, I < 1.52911 \times 10^{-41} \times e^{\left(\frac{T}{2.99919}\right)} + 13.8704 \\ 10.3, I \geq 1.52911 \times 10^{-41} \times e^{\left(\frac{T}{2.99919}\right)} + 13.8704 \end{cases}. \tag{7}$$

where $k = 11{,}079.30$, $a = -8.83 \times 10^{17}$, $b = -7.84 \times 10^{21}$, $E/R = 11{,}576.08$ K, and $T_0 = 298$ K. The units of reaction rate $k$ are μg $m^{-3}$ $min^{-1}$, the concentrations of $H^+$ and $Mn^{2+}$ are in mol/l, the surface area concentration $A$ is $nm^2/cm^3$, and the $SO_2$ mixing ratio is in ppbv.

**Water-soluble Mn measurement.** Each day, a $PM_{2.5}$ sample was collected on a 47-mm diameter quartz filter with a sampling flow rate of 5 l/min for 23.5 h. All quartz filters were pretreated before sample collection by baking at 450 °C in a muffle furnace for 6 h precluding possible existed contaminants. After cooling and after sampling, filters were wrapped in aluminum foil and stored in a refrigerator at −20 °C to prevent sample volatilization. For water-soluble Mn extraction, a quarter of the filter was shredded and ultrasonically dissolved by ultrapure water for 1 h. After extraction, nitric acid (68%, Beijing Institute of Chemical Reagents) was added to reach the final concentration of 2% nitric acid/sample solution. For total Mn extraction, another quarter of the filter was shredded and placed in a Teflon vessel to which a mixture of $HNO_3$ and HF in a ratio of 3:1 was added for microwave digestion. After cooling, the digested samples were transferred to cleaned centrifuge tubes diluted by 2% nitric acid, followed by weighting and filtering. The concentration of Mn in the prepared samples were measured by Inductively Coupled Plasma Mass Spectrometry.

**Observational dataset.** The model was evaluated using daily mass concentration data for water-soluble ions (e.g., $NH_4^+$, $SO_4^{2-}$, and $NO_3^-$) and TMIs (Fe and Mn) in $PM_{2.5}$ and gaseous species (e.g., $SO_2$, $NO_2$, and $O_3$), over the NCP during January 2015 and December 2016. Six prolonged haze episodes with daily $PM_{2.5}$ concentrations exceeding 150 μg $m^{-3}$ occurred over the NCP on January 2–5, 7–10, and 13–15, 2015, and December 2–4, 10–12, and 16–21, 2016.

Sounding data at Beijing, Xingtai, and Jinan, obtained from the Department of Atmospheric Science, University of Wyoming (http://weather.uwyo.edu/upperair/sounding.html), were used to identify the temperature inversion of the atmospheric boundary layer. Surface meteorological data over the NCP obtained from the National Climate Data Center integrated surface database (http://www.ncdc.noaa.gov/data-access/) were used to evaluate the performance of the meteorological simulations. The variables evaluated included the hourly temperature at 2 m (T2), RH at 2 m (RH2), wind speed at 10 m (WS10), and wind direction at 10 m (WD10). The statistical parameters included the correlation coefficient ($R$), mean bias (MB), normalized mean bias (NMB), and root mean square error.

**Model simulation.** WRF-Chem ver. 4.1.1[55,56] was used to investigate the sulfate formation mechanism. The simulation was performed on a domain covering the NCP, with 30 km horizontal resolution, 45 × 60 grid cells, and 23 vertical levels from the ground level to the maximum pressure of 50 hPa. Simulations were conducted from November 20 to December 31, 2016 and December 20, 2014 to January 31, 2015. The Noah land surface model and Yonsei University planetary boundary layer scheme were used to capture the land surface and boundary layer processes, respectively. The initial and boundary meteorological conditions were obtained from the National Centers for Environmental Prediction Final Analysis with a 6-h temporal resolution. The chemical mechanism used in this study was Carbon Bond Mechanism ver. Z[57] coupled with the Modal Aerosol Dynamics model for Europe[58] with the Secondary Organic Aerosol Model[59]. The original ISORROPIA model in WRF-Chem was replaced with the improved version (ISORROPIA II) in this study.

Anthropogenic emission data were from the Multi-resolution Emission Inventory for China model (http://www.meicmodel.org). Ammonia emissions were updated monthly at a 1 × 1 $km^2$ resolution based on our previous studies[60,61]. Recent studies showed that our results agreed well with satellite measurements[62] and inverse model results using ammonium wet deposition data[63,64].

Three simulations were performed. Experiment I used the standard WRF-Chem model, wherein sulfate is produced from gas phase oxidation of $SO_2$ by OH, and the in-cloud reaction of dissolved S(IV) with $H_2O_2$, $O_3$, $NO_2$, and $O_2$ catalyzed by

Fe(III) and Mn(II). In experiment II, manganese-catalyzed oxidation of $SO_2$ on the aerosol surface was implemented in the model used in experiment I. In experiment III, several recently highlighted reactions in the aerosol liquid phase (listed in Supplementary Table 2) was implemented in the model used in experiment II. Experiment III contains three sub-experiments corresponding to the three schemes for aqueous TMI-catalyzed $SO_2$ oxidation. In experiment III-a, the aqueous Mn(II) catalyzed oxidation in aerosol water was adopted. In experiment III-b, the aqueous Fe(III) catalyzed oxidation in aerosol water was adopted. In experiment III-c, the aqueous synergistic Fe(III)–Mn(II) catalyzed oxidation in aerosol water was adopted. In addition to the TMI-catalytic reactions, the reactions of S(IV) with $O_3$, $H_2O_2$, $NO_2$, and HCHO in aerosol water were also included in experiment III-a, III-b, and III-c. The model considered the influence of ionic strength on the oxidation of $SO_2$ by $H_2O_2$, $O_3$, and $O_2$ catalyzed by Mn(II) and Fe(III)-Mn(II). An integrated process rate analysis scheme[65] for sulfate was implemented in WRF-Chem to record the sulfate formation rate and calculate the contribution of each sulfate formation pathway.

In the model, the mass concentrations of Fe and Mn were scaled with the mass concentration of mineral aerosol. The mass fractions of Fe and Mn in the mineral aerosol were 3.5% and 0.3%, respectively. Only dissolved Fe and Mn in the oxidation states [Fe(III) and Mn(II) catalyze] S(IV) oxidation. The acids formed from anthropogenic and natural emissions dissolve Fe and Mn in airborne particles. The solubility of Fe and Mn was assumed to be 10% and 50%, respectively, based on global field observations[66–70]. Laboratory measurements in this study also showed that the solubility of Mn is about 50%. All of the dissolved Mn was assumed to be Mn(II), while the proportion of Fe(III) in the dissolved Fe was 10% and 90% during the day and at night, respectively. In experiments II and III, heterogeneous sulfate production, on the aerosol surface and in aerosol water, occurred only when the RH exceeded 35%.

**Model evaluation**. Supplementary Table 3 shows the meteorological prediction performance for 25 sites (marked by red triangles in Supplementary Fig. 20) over the NCP during haze episodes in December 2016 and January 2015. The predicted T2 agree well with the observations, with a correlation coefficient of 0.9 and MB of 0.3 °C. RH2 is underestimated slightly, with an MB of −11.0%. The simulated WS10 agrees reasonably well with the observations over the NCP, with an MB of 0.3 m s$^{-1}$. The model reasonably reproduces the temporal variation and magnitudes of meteorological variables, confirming the credibility of the meteorology simulation.

Figure 6d and Supplementary Fig. 15d compare the observed and WRF-Chem-simulated $SO_4^{2-}$ concentrations during the haze episodes in December 2016 and January 2015, respectively. In experiment I, $SO_4^{2-}$ is significantly underestimated, with a large NMB of −89.1% during the six haze episodes. In experiment II, the $SO_4^{2-}$ simulation is significantly improved, with a low NMB of −2.7%. In experiment III, the $SO_4^{2-}$ simulation is almost equivalent to Experiment II, with an NMB of 3.1%. In experiment II other species were evaluated. Supplementary Fig. 16 compares the observed and WRF-Chem-simulated concentrations of $NO_3^-$ and $NH_4^+$. The model generally captures the magnitude of $NO_3^-$ (average values 32.9 vs. 38.3 μg m$^{-3}$). The modeled $NH_4^+$ is slightly overestimated, with an NMB of 17.4%. Supplementary Fig. 17 compares the observed and WRF-Chem-simulated concentrations of Mn and Fe. The simulated Mn concentrations agree well with the observations, with a low NMB of 5.1%. The simulated Fe concentrations also agree well with the observations, with a low NMB of 2.3%. The simulated gaseous species ($SO_2$, $NO_2$, and $O_3$) are evaluated at nine sites over the NCP during the six haze episodes. Supplementary Fig. 18 compares the observed and WRF-Chem-simulated $SO_2$, $NO_2$, and $O_3$. The model generally captures the magnitude and spatial variation of $SO_2$, $NO_2$, and $O_3$. The observed and simulated mean $SO_2$ concentrations at the nine sites are 80.8 and 80.6 μg m$^{-3}$, respectively. The simulated $NO_2$ is slightly underestimated, with an NMB of −12.4%. The simulated $O_3$ agrees well with the observations, with a low NMB of 3.6%.

## Data availability
Source data are provided with this paper.

## Code availability
The source code for WRF-CHEM used in this paper is openly available from https://ruc.noaa.gov/wrf/wrf-chem/. The source code for sulfate simulation developed in this paper are available upon request from the corresponding author.

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

## Acknowledgements

W.W. acknowledges the support of the National Natural Science Foundation of China and National Key Research and Development Program of China (41822703, 2017YFC0209500). Y.S. and M.G. acknowledge National Natural Science Foundation of China (91644212, 91544227, 91844301). This work was partially supported by the National research program for key issues in air pollution control (DQGG-0103) and Beijing National Laboratory for Molecular Sciences (BNLMS-CXXM-202011). M.L. would like to thank Pianpian Chang and Jian Zhang for the assistance in micro-Raman experiments and single-particle analysis. W.W. is grateful to Prof. Jie Li and Prof. Dongsheng Ji for the helpful discussion and Li Deng for the ICP-MS measurements.

## Author contributions

W.W., Y.S., and M.G. conceived and led the studies. W.W. and M.L. performed chamber simulation and data analysis. M.L., Z.C., and Y.Z. performed micro-Raman experiments. M.L., J.C., J.H., G.T., Y.P., and W.L. performed in-field observation. T.W. improved source code and performed WRF-Chem simulation. W.W., M.L., T.W., Y.S., and M.G. wrote the paper. L.Z., Z.L., Z.X., C.P., C.L., Y.C., Y.S., T.Z., H.T. discussed the results and commented on the manuscript.

## Competing interests

The authors declare no competing interests.
