## [Peer Review File · Nature Communications]

REVIEWER COMMENTS

Reviewer #1 (Remarks to the Author):

The manuscript "Evidence of a new and dominate pathway for aerosol sulphate production during haze event" by Want et al presents new and interesting findings that deserve to be published in the prestigious journal Nature Communications.

The findings are certainly novel and the experiments conducted by the authors have been carefully designed.

The authors find that the sulphate oxidation in the Mn²⁺ containing aerosol particles proceeds more rapidly than previously reported for TMI catalysed sulphate oxidation by Mn²⁺. Their findings confirm a suspicion raised in an earlier study which found that the oxidation of SO₂ in a dust leachate containing a mixture of TMI including Fe, Ti, Mn and Cr was much more rapid than expected (Harris et al. 2012 *Atmos. Chem. Phys.*, 12, 4867–4884), namely, that the reaction rate coefficients of either individual TMIs or their composite effect must be reinvestigated. Most currently used rate coefficients were measured in the 1970s to 1990s and have not been reassessed since the review by Brand and Eldik (*Chem. Rev.* 1995, 95, 119-190).

I have only one significant concern regarding the manuscript which the authors should attempt to address. The experiments conducted clearly demonstrate that the seed material matters and that NaCl seed material results in faster oxidation compared to ammonium sulphate seed material. The attempts of the authors to justify the difference purely based on differences of the acidity of the seed material are unconvincing, since the pH of the aerosol changes rapidly once the reaction progresses. The authors should discuss this observation in greater depth keeping in mind the existing literature. The authors are not the first to observe that NaCl seed material results in faster Mn²⁺ catalyzed oxidation. Tursic et al. 2003 (*Atmospheric Environment Volume 37, Issue 19, June 2003, Pages 2589-2595*) investigated oxidation of dissolved SO₂ catalyzed by Mn(II) at pH3 under dark conditions and found the reaction progressed more rapidly on NaCl than on NaNO₃ seed material. While they attempted to explain the observed differences invoking a reaction mechanism involving Cl₂⁻ radical ions, that mechanism appears to be insufficient to explain the larger differences between the two seed materials observed in the current study, which does not involve an extremely acidic starting pH. On the other hand, Harris et al. 2012 (*Atmos. Chem. Phys.*, 12, 4619–4631) found rapid oxidation of SO₂ on NaOCl aerosol which produces HOCl in solution. This begs the question whether the simultaneous presence of Mn(II), O₂ and NaCl facilitates the rapid formation of HOCl in the aerosol liquid water and results in the formation of sulphate through two rather than just one extremely efficient oxidation pathways on this particular seed material. It should be noted, that the relative contribution of HOCl and TMI catalysed reactions to the product cannot be determined using isotopic techniques, as both oxidation processes result in similar isotope fractionation. The authors should also discuss whether or not the possible reaction rate enhancement due to the presence of NaCl has any atmospheric relevance in the wintertime haze scenario in China. While the supply of Cl to aqueous phase aerosol via uptake of HCl originating mostly from open burning of MSW would likely occur even under hazy conditions and far inland, marine aerosol may or may not contribute to the aerosol burden under stagnant conditions.

Minor comments:

L111 The observation that sulphate production rate increased with decreasing droplet diameter is interesting and the change in surface-area-to-volume-ratio which could favour more efficient uptake into the liquid may not be the only reason to blame. A lot of recent research has shown that acceleration of reaction kinetics in droplets relative to bulk solution is not uncommon (Yang et al. 2017, *Angew. Chem.* 2017, 129, 3616 –3619, Zhang et al. 2020 <https://dx.doi.org/10.1021/jasms.0c00057>). Unexpectedly rapid oxidation of SO₂ has been observed for several different reactions including the oxidation of SO₂ in small droplets without discernible oxidant (Hung et al. *Environ. Sci. Technol.* 2018, 52, 16, 9079–9086), which could possibly be explained by the spontaneous formation of H₂O₂ in shrinking droplets (Lee et al. 2019

PNAS September 24, 2019 116 (39) 19294-19298) and for the oxidation of SO₂ by H₂O₂ in simulated haze aerosol (Lui et al. PNAS January 21, 2020 117 (3) 1354-1359). There is increasing evidence, that the liquid air interface in droplets can catalyze or accelerate reactions and the possibility that this may be happening for the reactions proposed in the current study must be discussed.

L193 The authors should consider citing Li et al. 2020 Environ. Sci. Technol. 2020, 54, 5, 2626–2634. While the current study finds a somewhat larger contribution of TMI to the sulphate formation (72.5%±7.7%), than Li et al. 2020 (49 ± 10% contribution of TMI to the sulphate formation for a 2015 haze episode in Nanjing), both numbers cannot be explained without a much faster reaction rate for TMI catalysed oxidation compared to previously reported rate coefficient. Hence the current study plugs an important knowledge gap and helps in explaining ambient observations.

Baerbel Sinha

Reviewer #2 (Remarks to the Author):

Secondary sulfate is a key component of fine particles, and contributed greatly to the haze formation in China. The formation pathway of sulfate during the haze is still largely uncovered. The authors supposed Mn-catalyzed sulfate formation is a dominant pathway, rather than the liquid-phase oxidation on the basis of the chamber experiments and model simulations. The data shown here deepened our understanding on fine particle explosive growth during the haze formation. This manuscript could be published after major revision.

1 The transition metal (such as Fe and Mn)-catalyzed sulfate formation has been focused by many studies, and the mechanisms both of aqueous phase and particle surfaces were reported in detail, of which could not be considered as "a new pathway". I thus suggested that the thesis should be changed "Evidence of a dominant pathway for aerosol sulfate production during haze events"

2 Fe often co-existed with Mn in nature. During the past years, Fe metal was more focused on the heterogeneous reaction as compared to Mn metal due to high concentration in the atmosphere and multi-source. I am concern that what a role Fe element play under the same condition with Mn metal. Could the author compare the sulfate formation rate involved Fe and Mn using the Chamber experiment. In the WRF-Chem using in this manuscript, they only compared Mn-catalyzed to Fe/Mn in aerosol water. They attributed the Fe/Mn in liquid to "the others", which totally take up 0.3%, together with in-cloud oxidation and reactions of SO₂ with O₃, NO₂, O₂ catalyzed by Mn. Could you show a result of Fe-catalyzed alone, and how to contribute to sulfate formation?

3 Generally, the author supposed Mn-catalyzed is a main pathway, but they should supply more proof. For example, to build a relationship between field-measurement and chamber experiment. Oxygen and sulfur isotopic method could solve this problem. It is better if they could identify Mn concentrations, and source (dust or coal combustion), and build a touch between sulfate and soluble Mn and/or Fe.

Reviewer #3 (Remarks to the Author):

This paper adds a new hypothesis on the chemical mechanism leading to the production of fine particle sulfate in Beijing (or North China Plane) during winter heavy pollution events; another addition to the growing list of possible mechanism that have been proposed in the last two years or so.

The authors provide arguments discounting the other mechanism and propose instead a mechanism that depends on chemistry occurring at the surface of liquid drops and involves Mn^{2+} . They base their conclusion that this is an important mechanism on chamber and lab experiments and the implementation in a chemical model that produces reasonable agreement with observations.

The paper is well written, but a number of clarifications are needed. What exactly is meant by these chemical reactions occurring on deliquesced aerosol surfaces. A schematic would be helpful. I assume this means that there is a liquid drop (haze particle, ie $PM_{2.5}$, not cloud or fog) containing all the dissolved species, such as Mg^{2+} , and that the SO_2 is absorbed at this surface and the chemical reactions happen in that liquid surface layer. Then these species are dispersed throughout the liquid drop volume. Is that the conceptual model?

Please explain in detail exactly what is the difference between this Mn^{2+} catalyzed surface chemistry vs bulk Mn^{2+} catalyzed chemistry, and why the surface chemistry leads to orders of magnitude sulfate production.

The TEM images of Fig 1c are difficult to understand. Please state exactly how they show the Mn-catalytic reaction.

The evidence for a surface driven reaction seems mainly to be from the Raman single particle analysis that showed increase sulfate mass scales with inverse of particle diameter. Is this true? If so, how does a wet drop sitting on a particle surface, the conditions of the Raman experiment, affect this observation. Eg, how was the particle diameter determined, what is the surface area of a drop sitting on a surface?

Finally, and of the most importance, if this mechanism is a major route than most of the $PM_{2.5}$ sulfate in ambient particles during winter haze must be internally mixed with Mn^{2+} (or just Mn). The authors seem to have the capability to do single particle analysis, ie Raman or TEM. Or maybe they could collaborate with researchers who have the instrument capable of making this measurement. Alternatively, one could use a cascade impactor to compare the size distribution of sulfate to Mn. I believe size distribution data during these events are available, (published). The authors should check if these data exist and how they support this proposed mechanism, example, maybe one could see if there is sufficient Mn in the size ranges where most the sulfate is found to make this a viable mechanism, based on their chemical model. Since both sulfate and Mn are nonvolatile, sampling issues should not confound the experiment. This is a unique situation, the authors can directly test if their hypothesis is valid (internally sulfate – Mn mixing is a necessary, but not sufficient condition), all they have to do is make the measurement or get the data somehow.

Point-by-Point Response

Please find our detailed point-by-point response to reviewers' comments below. The original comments are in black, our responses are in blue, and the contents added to the revised manuscript are *in italicized blue*.

Reviewer #1 (Remarks to the Author):

The manuscript “Evidence of a new and dominate pathway for aerosol sulphate production during haze event” by Want et al presents new and interesting findings that deserve to be published in the prestigious journal Nature Communications. The findings are certainly novel and the experiments conducted by the authors have been carefully designed.

Response: We thank the reviewer for the positive comments regarding our work.

The authors find that the sulphate oxidation in the Mn²⁺ containing aerosol particles proceeds more rapidly than previously reported for TMI catalysed sulphate oxidation by Mn²⁺. Their findings confirm a suspicion raised in an earlier study which found that the oxidation of SO₂ in a dust leachate containing a mixture of TMI including Fe, Ti, Mn and Cr was much more rapid than expected (Harris et al. 2012 Atmos. Chem. Phys., 12, 4867–4884), namely, that the reaction rate coefficients of either individual TMIs or their composite effect must be reinvestigated. Most currently used rate coefficients were measured in the 1970s to 1990s and have not been reassessed since the review by Brand and Eldik (Chem. Rev. 1995, 95, 119-190).

Response: We completely agree with your comments. Yes, most of the current research using the kinetics of TMIs catalytic SO₂ oxidation reaction were measured in bulk solution in last century, and their roles should be reassessed especially under close to atmospheric conditions. The findings derived from Harris et al. (2012) strongly supported our results that the reaction rate of the TMI-catalytic reaction could be much faster than expected, we cited this reference in our manuscript.

Page 4, line 97

Harris et al.¹ also detected an unexpected rapid SO₂ oxidation rate in a dust leachate, which contained a mixture of TMI (Fe, Ti, Mn, and Cr) rather than the iron-only catalytic pathway. In our study, compared to Mn-catalytic reaction, there was no noticeable sulfate formation observed through Fe-catalytic pathway in the chamber experiment (Supplementary Fig. 3), and the reaction rate of the Fe-Mn-catalytic reaction did not show any synergism enhancement effect. This evidence indicated that the role of

individual TMIs or their composite effect to sulfate formation need further reinvestigation under close to real atmosphere conditions.

I have only one significant concern regarding the manuscript which the authors should attempt to address. The experiments conducted clearly demonstrate that the seed material matters and that NaCl seed material results in faster oxidation compared to ammonium sulphate seed material. The attempts of the authors to justify the difference purely based on differences of the acidity of the seed material are unconvincing, since the pH of the aerosol changes rapidly once the reaction progresses. The authors should discuss this observation in greater depth keeping in mind the existing literature. The authors are not the first to observe that NaCl seed material results in faster Mn²⁺ catalyzed oxidation. Tursic et al. 2003 (Atmospheric Environment Volume 37, Issue 19, June 2003, Pages 2589-2595) investigated oxidation of dissolved SO₂ catalyzed by Mn(II) at pH3 under dark conditions and found the reaction progressed more rapidly on NaCl than on NaNO₃ seed material. While they attempted to explain the observed differences invoking a reaction mechanism involving Cl₂⁻ radical ions, that mechanism appears to be insufficient to explain the larger differences between the two seed materials observed in the current study, which does not involve an extremely acidic starting pH.

Response: We thank the reviewer for pointing out the possible Cl⁻ catalytic effect in our system. Tursic et al. (2003) conducted their experiments in bulk solution under relatively a low ionic strength and high acidity condition, and they found that Cl⁻ catalytic effect would be promoted under lower ionic strength condition and strongly inhibited under higher ionic strength condition ($I > 0.02 \text{ mol L}^{-1}$). In our experiment system, the ionic strength of NaCl seed particle could reach at least 2.8 mol L^{-1} (90% RH). These different conditions indicate that the Cl⁻ catalytic effect might be less effective in our experimental system. To examine this possibility scientifically, we conducted experiments using NaNO₃ as the seed particle. We stated the results in the next response.

In our study, the particle acidity changes rapidly during the reaction. The pH values in data analysis were calculated by the thermodynamic model E-AIM model IV in each step, by considering the content of sulfate and ammonia changes throughout the reaction process.

Page 5, line 116

In the Mn²⁺ catalytic aqueous phase reaction, Tursic et al.² found that substituting NaCl for NaNO₃ as the seed material could lead to a faster reaction under acidic (pH=3) and dark conditions due to the formation of Cl₂⁻ radical ions.

On the other hand, Harris et al. 2012 (Atmos. Chem. Phys., 12, 4619–4631) found rapid oxidation of SO₂ on NaOCl aerosol which produces HOCl in solution. This begs the question whether the simultaneous presence of Mn(II), O₂ and NaCl facilitates the rapid formation of HOCl in the aerosol liquid water and results in the formation of sulphate through two rather than just one extremely efficient oxidation pathways on this particular seed material. It should be noted, that the relative contribution of HOCl and TMI catalysed reactions to the product cannot be determined using isotopic techniques, as both oxidation processes result in similar isotope fractionation. The authors should also discuss whether or not the possible reaction rate enhancement due to the presence of NaCl has any atmospheric relevance in the wintertime haze scenario in China. While the supply of Cl to aqueous phase aerosol via uptake of HCl originating mostly from open burning of MSW would likely occur even under hazy conditions and far inland, marine aerosol may or may not contribute to the aerosol burden under stagnant conditions.

Response: We thank the reviewer for raising this important question as to whether the simultaneous presence of Mn, O₂, and NaCl seed particles would facilitate formation of HOCl and cause rapid sulfate formation (Harris et al., 2012).

Thus, we conducted additional chamber experiments by substituting NaNO₃ for NaCl under the same conditions. The result showed that the reaction rate of NaNO₃ seed experiments were consistent with the equations (Eq. 1 in main text and M1–M4 in method section) derived from NaCl and (NH₄)₂SO₄ experiments. This demonstrated that there was no detectable reaction rate enhancement in the NaCl system compared to the NaNO₃ system. Hence, HOCl was less likely to form or play a vital role in the simultaneous presence of Mn, O₂, and NaCl. We added the discussion of HOCl to our manuscript. We also would like to thank the reviewer for the thoughtful suggestion to expand the implication of our work. As described above, the increased content of NaCl is mainly through lower aerosol acidity, which then leads to a more rapid sulfate formation rate.

Page 5, line 118

Moreover, Harris et al.³ found that SO₂ was rapidly oxidized on the NaOCl aerosol which produced HOCl in solution. To examine whether the simultaneous presence of Mn²⁺, O₂, and Cl⁻ in aerosol liquid water would facilitate HOCl formation, chamber experiments were also conducted under the NaNO₃ seed condition. As depicted in Supplementary Fig. 4, there was no clear reaction rate decrease in the NaNO₃ seed reaction.

Supplementary Fig. 4. Reaction rate of a sodium nitrate seed particle at 298 K compared to model simulation (M1–M4). In chamber experiments, sodium nitrate particles containing different concentration of 0.41% Mn^{2+} were exposed to SO_2 (116 ppb) and NH_3 (84 ppb) under RH 89%. There was no obvious reaction rate decrease in $NaNO_3$ seed reaction compared to the model (Fig. 3a, Method M1–M4).

Minor comments:

L111 The observation that sulphate production rate increased with decreasing droplet diameter is interesting and the change in surface-area-to-volume-ratio which could favour more efficient uptake into the liquid may not be the only reason to blame. A lot of recent research has shown that acceleration of reaction kinetics in droplets relative to bulk solution is not uncommon (Yang et al. 2017, Angew. Chem. 2017, 129, 3616–3619, Zhang et al. 2020 <https://dx.doi.org/10.1021/jasms.0c00057>). Unexpectedly rapid oxidation of SO_2 has been observed for several different reactions including the oxidation of SO_2 in small droplets without discernible oxidant (Hung et al. Environ. Sci. Technol. 2018, 52, 16, 9079–9086), which could possibly be explained by the spontaneous formation of H_2O_2 in shrinking droplets (Lee et al. 2019 PNAS September 24, 2019 116 (39) 19294-19298) and for the oxidation of SO_2 by H_2O_2 in simulated haze aerosol (Lui et al. PNAS January 21, 2020 117 (3) 1354-1359).

There is increasing evidence, that the liquid air interface in droplets can catalyze or accelerate reactions and the possibility that this may be happening for the reactions proposed in the current study must be discussed.

Response: We thank the reviewer for suggesting these possible explanations as to why a surface reaction obtains a higher reaction rate than an aqueous reaction. Yan et al. (2017) found acceleration of two-phase reaction in the microdroplet interface without the use of phase-transfer catalysts, and Zhang et al. (2020) showed a four-order of magnitude larger reaction rate of the Girard's T reaction by droplet assisted ionization with microdroplets compared to bulk solution. Hung et al. (2018) proposed sulfate formation in the reaction of sulfurous acid microdroplet and oxygen, without additional oxidants, and the finding of spontaneously H₂O₂ production in the atomized bulk water microdroplets by Lee et al. (2019) may shed light on this phenomenon. Also, Liu et al. (2020) observed an enhanced sulfate formation rate by H₂O₂ aqueous oxidation in aerosol particles with high ionic strength. These findings demonstrated that there may be a massive gap in the reaction rate between the air-liquid interface and bulk solution, which is consistent with our work.

In our experimental system, monodisperse neutral particles were produced and added into the chamber system; we cannot observe sulfate formation in the absence of Mn. This phenomenon demonstrated that the mechanism of the air-liquid interface rate enhancement effect may differ greatly in a distinct reaction system. We added further discussion on the surface reaction and differences between the Mn-catalytic heterogeneous phase reaction and the aqueous phase reaction in our revised manuscript.

Page 8, line 174

The Mn-catalyzed aqueous phase oxidation of S(IV) has been extensively investigated and the critical oxidation process is the reaction between Mn(III) and hydrogen sulfite or its complex with Mn(II). Then SO₃⁻ radicals can react with dissolved O₂ to generate SO₅⁻ radicals, which can oxidize Mn(II) to regenerate Mn(III). The bulk manganese catalytic reaction is the ion-ion reaction, in which rate constant decreases dramatically with increasing ionic strength, influenced by the primary kinetic salt effect. The enhanced reaction rate for Mn-catalytic reaction on the aerosol surface compared to aqueous reaction is mainly attributed to several differences: the difference between the ion-neutral molecule and ion-ion reactions, which perform differently under high ionic strength; the different reaction space; and the surface of the aerosol and bulk solution. The aerosol water content is scarce compared with cloud water content, and the low pH values limit the solubility of SO₂. The surface-area-to-volume ratio increases with the decreasing of the aerosol diameter, making the chemistry on the surface more important than in the bulk phase. Furthermore, Yan et al.⁴ and Zhang et al.⁵ found that reaction occurring in microdroplets obtained a higher rate than the same reaction in bulk solution. Hung et al.⁶ proposed sulfate formation in the reaction of sulfurous acid microdroplet and oxygen, without additional oxidants, and the finding of spontaneously H₂O₂ production in

the atomized bulk water microdroplets by Lee et al.⁷ may shed light on this phenomenon. These findings demonstrated that there may be a massive gap in the reaction rate between air–liquid interface and bulk solution, and the mechanism of air–liquid interface rate enhancement effect may differ greatly in a distinct reaction system.

L193 The authors should consider citing Li et al. 2020 Environ. Sci. Technol. 2020, 54, 5, 2626–2634. While the current study finds a somewhat larger contribution of TMI to the sulphate formation (72.5%+7.7%), than Li et al. 2020 (49 ± 10% contribution of TMI to the sulphate formation for a 2015 haze episode in Nanjing), both numbers cannot be explained without a much faster reaction rate for TMI catalysed oxidation compared to previously reported rate coefficient. Hence the current study plugs an important knowledge gap and helps in explaining ambient observations.

Response: We thank the reviewer for providing additional material to support the importance of the TMI-catalytic SO₂ oxidation pathway in sulfate formation in China. We cited Li et al. (2020) in our revised manuscript, which supplied solid field observation evidence that the reaction rate of TMI-catalyzed oxidation should be more rapid than the reported rate coefficient measured in bulk solution. Thus, in light of your thoughtful advice, we added some remarks about stable sulfur isotope results.

Page 13, line 291

Consistently, stable sulfur isotope results supplied solid field observation evidence that TMI-catalytic reaction can be the dominant pathway for sulfate formation. TMIs contributed 49% ± 10% to sulfate formation in winter 2015 in Nanjing⁹.

References

1. Harris E, Sinha B, Foley S, Crowley JN, Borrmann S, Hoppe P. Sulfur isotope fractionation during heterogeneous oxidation of SO₂ on mineral dust. *Atmospheric Chemistry and Physics* **12**, 4867-4884 (2012).
2. Turšič J, Grgić I, Podkrajšek B. Influence of ionic strength on aqueous oxidation of SO₂ catalyzed by manganese. *Atmospheric Environment* **37**, 2589-2595 (2003).
3. Harris E, Sinha B, Hoppe P, Foley S, Borrmann S. Fractionation of sulfur isotopes during heterogeneous oxidation of SO₂ on sea salt aerosol: a new tool to investigate non-sea salt sulfate production in the marine boundary layer. *Atmospheric Chemistry and Physics* **12**, 4619 (2012).
4. Yan X, Cheng H, Zare RN. Two-phase reactions in microdroplets without the use

- of phase-transfer catalysts. *Angewandte Chemie* **129**, 3616-3619 (2017).
5. Zhang Y, Apsokardu MJ, Kerecman DE, Achtenhagen M, Johnston MV. Reaction Kinetics of Organic Aerosol Studied by Droplet Assisted Ionization: Enhanced Reactivity in Droplets Relative to Bulk Solution. *Journal of the American Society for Mass Spectrometry*, (2020).
 6. Hung HM, Hsu MN, Hoffmann MR. Quantification of SO₂ Oxidation on Interfacial Surfaces of Acidic Micro-Droplets: Implication for Ambient Sulfate Formation. *Environ Sci Technol* **52**, 9079-9086 (2018).
 7. Lee JK, *et al.* Spontaneous generation of hydrogen peroxide from aqueous microdroplets. *Proceedings of the National Academy of Sciences* **116**, 19294-19298 (2019).
 8. Liu T, Clegg SL, Abbatt JPD. Fast oxidation of sulfur dioxide by hydrogen peroxide in deliquesced aerosol particles. *Proceedings of the National Academy of Sciences* **117**, 1354-1359 (2020).
 9. Li J, *et al.* Stable Sulfur Isotopes Revealed a Major Role of Transition-Metal-Ion Catalyzed SO₂ Oxidation in Haze Episodes. *Environmental Science & Technology* (2020).

Reviewer #2 (Remarks to the Author):

Secondary sulfate is a key component of fine particles, and contributed greatly to the haze formation in China. The formation pathway of sulfate during the haze is still largely uncovered. The authors supposed Mn-catalyzed sulfate formation is a dominant pathway, rather than the liquid-phase oxidation on the basis of the chamber experiments and model simulations. The data shown here deepened our understanding on fine particle explosive growth during the haze formation. This manuscript could be published after major revision.

Response: We thank the reviewer for commending of our work. A detailed response to the concerns is addressed as follows.

1 The transition metal (such as Fe and Mn)-catalyzed sulfate formation has been focused by many studies, and the mechanisms both of aqueous phase and particle surfaces were reported in detail, of which could not be considered as “a new pathway”. I thus suggested that the thesis should be changed “Evidence of a dominant pathway for aerosol sulfate production during haze events”

Response: Accepted. The title was changed in the revised manuscript.

Title: *Evidence of a dominant pathway for aerosol sulfate production during haze events*

2 Fe often co-existed with Mn in nature. During the past years, Fe metal was more focused on the heterogeneous reaction as compared to Mn metal due to high concentration in the atmosphere and multi-source. I am concern that what a role Fe element play under the same condition with Mn metal. Could the author compare the sulfate formation rate involved Fe and Mn using the Chamber experiment. In the WRF-Chem using in this manuscript, they only compared Mn-catalyzed to Fe/Mn in aerosol water. They attributed the Fe/Mn in liquid to “the others”, which totally take up 0.3%, together with in-cloud oxidation and reactions of SO₂ with O₃, NO₂, O₂ catalyzed by Mn. Could you show a result of Fe-catalyzed alone, and how to contribute to sulfate formation?

Response: We thank the reviewer for this insightful comment; the additional discussion on Fe and Fe/Mn could enrich our manuscript. We supplemented experimental analysis and model simulation on the role of the Fe element in sulfate formation.

Firstly, we carried out Fe and Fe/Mn catalytic SO₂ oxidation chamber experiments under the same conditions as the Mn-catalytic reaction, and the

results are shown in Supplementary Fig. 3. In the Fe-catalytic reaction, no noticeable sulfate formation was observed, indicating that Fe did not have the heterogeneous reaction property that Mn did, and the aqueous-phase reaction would be strongly inhibited by high ionic strength. In the Fe–Mn catalytic reaction, the reaction rate did not show a noticeable synergism enhancement; moreover, excessive addition of Fe would increase aerosol acidity due to its hydrolysis and lower reaction rate.

Supplementary Fig. 3. Temporal profiles of the particle mass concentration measured under different conditions at 298 K. In the chamber experiments, NaCl particles containing different concentration of Fe^{3+} and Mn^{2+} were exposed to SO_2 (116 ppb) and NH_3 (84 ppb) under RH 89%.

Secondly, in model simulation, we conducted three scenarios to investigate the effectiveness of the three mechanisms of TMI-catalyzed SO_2 oxidation in aerosol water, including Mn(II), Fe(III), and the synergistic Fe(III)–Mn(II). The contributions of Mn(II) catalyzed, Fe(III) catalyzed, and the synergistic Fe(III)–Mn(II) catalyzed oxidation to sulfate concentration were $0.09\% \pm 0.05\%$, $0.02\% \pm 0.02\%$, and $0.05\% \pm 0.08\%$, respectively. The low concentration of Fe^{3+} in the moderately acidic aerosol water is not conducive to the catalytic reactions of iron. In addition, the rates of the Mn(II) catalyzed and the synergistic Fe(III)–Mn(II) catalyzed oxidation in liquid phase decreased exponentially with the increasing ionic strength (Seinfeld & Pandis., 2016; Martin et al., 1987) and were suppressed by the high ionic strength in aerosol water.

Page 5, line 99

In our study, compared to the Mn-catalytic reaction, there was no noticeable sulfate formation observed through the Fe-catalytic pathway in the chamber experiment

(Supplementary Fig. 3), and the reaction rate of the Fe–Mn-catalytic reaction did not show any synergism enhancement effect. This evidence indicated that the role of individual TMIs or their composite effect to sulfate formation need further reinvestigation under close to real atmosphere conditions.

Page 28, line 501

Experiment III contains three sub-experiments corresponding to the three schemes for aqueous TMI-catalyzed SO₂ oxidation. In experiment III-a, the aqueous Mn(II) catalyzed oxidation in aerosol water was adopted. In experiment III-b, the aqueous Fe(III) catalyzed oxidation in aerosol water was adopted. In experiment III-c, the aqueous synergistic Fe(III)–Mn(II) catalyzed oxidation in aerosol water was adopted. In addition to the TMI catalytic reactions, the reactions of S(IV) with O₃, H₂O₂, NO₂, and HCHO in aerosol water were also included in experiment III-a, III-b, and III-c.

Page 12, line 275

The TMIs-catalyzed SO₂ oxidation reactions in aerosol water also contribute little to sulfate. Three schemes of Mn(II) catalyzed-, Fe(III) catalyzed-, and the synergistic Fe(III)–Mn(II)-catalyzed oxidation were tested, and their contributions were 0.09% ± 0.05%, 0.02% ± 0.02%, and 0.05% ± 0.08%, respectively. The low concentration of Fe³⁺ in the moderately acidic aerosol water is not conducive to the catalytic reactions of iron. In addition, the rates of the Mn(II)-catalyzed and the synergistic Fe(III)–Mn(II)-catalyzed oxidation in liquid phase decreased exponentially with increasing ionic strength and were suppressed by high ionic strength in aerosol water (30–50 mol L⁻¹; Supplementary Fig. 19d).

3 Generally, the author supposed Mn-catalyzed is a main pathway, but they should supply more proof. For example, to build a relationship between field-measurement and chamber experiment. Oxygen and sulfur isotopic method could solve this problem. It is better if they could identify Mn concentrations, and source (dust or coal combustion), and build a touch between sulfate and soluble Mn and/or Fe.

Response: We thank the reviewer for this thoughtful concern, and we agree that more evidence would better support our supposed mechanism. We added several field measurement phenomena to strengthen our manuscript listed in following.

- (1) As shown in Supplementary Fig. 10, we analyzed the correlation between Mn and sulfate concentration in the Northern China Plain (Baoding, Tianjin, Beijing, Tangshan, Dezhou, and Xinxiang) from Dec 1st to 22nd, 2016. The concentration of Mn was within the range of 21.3–91.9 ng m⁻³, 14.1–105.6 ng m⁻³, 14.5–107.6 ng m⁻³, 20.4–327.0 ng m⁻³, 31.3–92.7 ng m⁻³ and 29.1–123.6 ng m⁻³ respectively. In all sampling sites, the mass concentration of

Mn showed similar variation trend as that of sulfate, with the exception of a small part of the sulfate rise and drop. Using ambient PM_{2.5} chemical component data and positive matrix factorization method, source apportionment results showed that more than 80% Mn were emitted from coal combustion during haze events (Supplementary Fig. 14).

- (2) To further examine the correlation between water-soluble Mn and sulfate, we conducted field observations in two cities in the Northern China Plain (Baoding and Tianjin) from Jan 1st to 12th, 2015 during the same period of model simulation. In each day, a PM_{2.5} sample was collected on a 47-mm diameter quartz filter with a sampling flow rate of 5 L/min for 23.5 h. For water-soluble Mn extraction, a quarter of the filter was shredded and ultrasonically dissolved by ultrapure water for 1 h. For total Mn extraction, another quarter of the filter was shredded and placed in a Teflon vessel; a mixture of HNO₃ and HF at a ratio of 3:1 was added for microwave digestion. The concentration of Mn in prepared samples was measured by inductively coupled plasma mass spectrometry (ICP-MS). As shown in Fig. 5, compared with total Mn, water-soluble Mn precisely tracked the sulfate variation trend in the observation period. Consistent with chamber experiments, these results demonstrate that water-soluble Mn plays a key role in sulfate formation. Additionally, the solubility of Mn ranges from 21.1% to 79.8% with an average of 47.9% ± 17.0%, indicating that the Mn solubility of 50% used in the model simulation was appropriate. Based on the added field observation results, we optimized the date of haze episodes and the time and space range of meteorological evaluation of WRF-Chem focusing on the North China Plain in model simulation.
- (3) Furthermore, we measured the size distribution of Mn and sulfate sampled in different heavy pollution periods (with averaged PM_{2.5} concentrations of 250, 310, 300 µg m⁻³ from left to right in Supplementary Fig. 11, respectively) in Beijing. We found that Mn and sulfate exhibited similar size distribution in accumulation mode, indicating the contribution of Mn in the sulfate formation process.

Recently, several stable sulfur isotope (³⁴S) results measured in China have been published. *TMI-catalytic reaction can be the dominant pathway for sulfate formation.* We have referenced these past studies and related discussion in our revised manuscript.

Page 9, line 210

Several in-field observations were conducted to build a relationship between the proposed pathway and haze episodes. As shown in Supplementary Fig. 10, we analyzed

the correlation between Mn and sulfate concentration in the Northern China Plain (Baoding, Tianjin, Beijing, Tangshan, Dezhou and Xinxiang) from Dec 1st to 22nd, 2016. The concentration of Mn was within the range of 21.3–91.9 ng m⁻³, 14.1–105.6 ng m⁻³, 14.5–107.6 ng m⁻³, 20.4–327.0 ng m⁻³, 31.3–92.7 ng m⁻³, and 29.1–123.6 ng m⁻³ respectively. In all sampling sites, mass concentration of Mn showed a similar variation trend as that of sulfate, with the exception of a small part of the sulfate rise and drop. Furthermore, we conducted water-soluble Mn concentration in the NCP during winter. Compared with total Mn, water-soluble Mn precisely tracked all of the sulfate variation trends in the observation period. (Fig. 5). Since these results were consistent with our prior experiments, they provided solid evidence that dissolved Mn played a key role in sulfate formation. The size distribution of Mn and sulfate content sampled in wintertime Beijing in 2013, 2014, and 2016 showed a similar variation trend in accumulation mode particles during different polluted periods (Supplementary Fig. 11), implying that the formation process of sulfate could be related closely to Mn.

Page 10, line 232

Using ambient PM_{2.5} chemical component data and positive matrix factorization method, source apportionment results showed that more than 80% Mn were emitted from coal combustion during haze events (Supplementary Fig. 14).

Supplementary Fig. 10. Temporal variation of sulfate and Mn concentration in PM_{2.5} at different sampling sites in NCP (Baoding, Tianjin, Beijing, Tangshan, Dezhou, and Xinxiang) in December 2016. Blue circles refer to Mn, Red squares refer to sulfate.

Supplementary Fig. 14. Source contributions of Mn in North China Plain from Dec. 1st to 22nd, 2016 by using ambient $PM_{2.5}$ chemical compositions and positive matrix factorization method. The sources include coal combustion, industrial process, soil dust, biomass burning and vehicle emission.

Fig. 5. Temporal variation of water-soluble Mn, total Mn, and sulfate concentration in $PM_{2.5}$ in North China Plain (Baoding and Tianjin) from Jan. 1st to Jan. 12th, 2015. (a) represents Baoding and (b) refers to Tianjin. Metal ions and sulfate concentration were measured by inductively coupled plasma mass spectrometry and ion chromatography, respectively.

Supplementary Fig. 11. Size distribution of sulfate and Mn in aerosol in Beijing. Sampling period: (a) Jan. 28th–29th, 2013, (b) Feb. 24th–26th, 2014, and (c) Dec. 19th–20th, 2016. The red line represents sulfate content; the blue line represents Mn content.

Page 13, line 288

Consistently, stable sulfur isotope results supplied solid field observation evidence that the TMI-catalytic reaction can be the dominant pathway for sulfate formation. TMIs contributed $49\% \pm 10\%$ for sulfate formation in winter 2015 in Nanjing¹. (Line 250)

Method

Water-soluble Mn measurement

Each day, a PM_{2.5} sample was collected on a 47-mm diameter quartz filter with a sampling flow rate of 5 L/min for 23.5 h. All quartz filters were pretreated before sample collection by baking at 450°C in a muffle furnace for 6 h precluding possible existed contaminants. After cooling and after sampling, filters were wrapped in aluminum foil and stored in a refrigerator at -20°C to prevent sample volatilization. For water-soluble Mn extraction, a quarter of the filter was shredded and ultrasonically dissolved by ultrapure water for 1 h. After extraction, nitric acid (68%, Beijing Institute of Chemical Reagents) was added to reach the final concentration of 2% nitric acid/sample solution. For total Mn extraction, another quarter of the filter was shredded and placed in a Teflon vessel to which a mixture of HNO₃ and HF in a ratio of 3:1 was added for microwave digestion. After cooling, the digested samples were transferred to cleaned centrifuge tubes diluted by 2% nitric acid, followed by weighting and filtering. The concentration of Mn in the prepared samples were measured by Inductively Coupled Plasma Mass Spectrometry (ICP-MS).

Reference

1. Li J, et al. Stable Sulfur Isotopes Revealed a Major Role of Transition-Metal-Ion Catalyzed SO₂ Oxidation in Haze Episodes. *Environmental Science & Technology* (2020).
2. Seinfeld, J. H. & Pandis, S. N. *Atmospheric chemistry and physics: from air pollution to climate change.* (John Wiley & Sons, 2016).
3. Martin, L. R. & Hill, M. W. The Iron Catalyzed Oxidation of Sulfur - Reconciliation of the Literature Rates. *Atmos. Environ.* 21, 1487-1490 (1987).

Reviewer #3 (Remarks to the Author):

This paper adds a new hypothesis on the chemical mechanism leading to the production of fine particle sulfate in Beijing (or North China Plane) during winter heavy pollution events; another addition to the growing list of possible mechanism that have been proposed in the last two years or so.

The authors provide arguments discounting the other mechanism and propose instead a mechanism that depends on chemistry occurring at the surface of liquid drops and involves Mn^{2+} . They base their conclusion that this is an important mechanism on chamber and lab experiments and the implementation in a chemical model that produces reasonable agreement with observations.

Response: We are grateful to the reviewer for carefully reviewing and accurately summarizing our manuscript. Please find our responses below.

The paper is well written, but a number of clarifications are needed. What exactly is meant by these chemical reactions occurring on deliquesced aerosol surfaces. A schematic would be helpful. I assume this means that there is a liquid drop (haze particle, ie PM2.5, not cloud or fog) containing all the dissolved species, such as Mg^{2+} , and that the SO_2 is absorbed at this surface and the chemical reactions happen in that liquid surface layer. Then these species are dispersed throughout the liquid drop volume. Is that the conceptual model?

Response: We thank the reviewer for raising the point to make the manuscript more accessible to readers, and the reviewer indeed accurately described the process of Mn-catalytic surface reaction in our work. We added a schematic in the revised manuscript (Fig. 4), and a description of the conceptual model in the revised manuscript. The detailed reaction mechanism is described in the following response.

Page 7, line 163

Schematically, SO_2 is firstly absorbed at the surface layer of droplets containing dissolved Mn^{2+} and seed species (ammonium sulfate or sodium chloride in this work). Then, the Mn-catalytic reaction rapidly occurred at the surface layer, and the formed sulfate is finally dispersed throughout the liquid phase (Fig. 4).

Fig. 4. Schematic of Mn-catalytic oxidation of SO_2 on the aerosol surface and in the aqueous phase.

Please explain in detail exactly what is the difference between this Mn^{2+} catalyzed surface chemistry vs bulk Mn^{2+} catalyzed chemistry, and why the surface chemistry leads to orders of magnitude sulfate production.

Response: We thank again the reviewer for thoroughly reviewing this manuscript.

The Mn-catalyzed aqueous phase oxidation of S(IV) has been extensively investigated in the last century. The mechanism was proposed by several groups, and the critical oxidation process is the reaction between Mn (III) and hydrogen sulfite or its complex with Mn(II). Then, $\text{SO}_3^{\bullet-}$ radicals can react with dissolved O_2 to generate SO_5^{\ominus} radicals, which can oxidize Mn (II) to regenerate Mn (III). The bulk manganese catalytic reaction is the ion-ion reaction, similar to the aqueous iron catalytic reaction. The reaction rate constant decreases dramatically with increasing ionic strength, influenced by the primary kinetic salt effect (Hermann et al., 2003).

The mechanism for the Mn-catalyzed aqueous phase oxidation of S(IV) according to the literature (Hermann et al., 2003; Deguillaume et al., 2005; Fronaeus et al., 1998; Berglund et al., 1995) is:

According to our experimental results, the sulfate formation rate was correlated with the particle surface area, aerosol acidity, Mn^{2+} concentration, and SO_2 concentration. The Mn-catalytic redox reaction on aerosol surfaces is proposed. In this Mn-catalyzed heterogeneous reaction, Mn (III) is the intermediate product that could oxidize S(IV), which can directly oxidize SO_2 and O_2 in the liquid surface layer. Due to the high value of the hydrolysis equilibrium constant of Mn^{3+} , Mn (III) mainly exists in the forms of $Mn(OH)_2^-$ and $Mn(OH)^{2-}$, and their concentrations are correlated with the aerosol phase acidity and Mn(II) concentration. Under high ionic strength conditions, high electrolyte concentrations may accelerate the reaction rate in ion–neutral molecular reactions by forming an association or ion pair as a new activation center (Herrmann et al., 2003). The energy barrier of these intermediate products might be influenced by temperature changes, leading to a temperature-related enhancement effect of the ionic strength threshold.

The enhanced reaction rate for Mn-catalytic reaction on the aerosol surface compared to aqueous reaction mainly attribute to several differences: the difference between the ion–neutral molecule and ion–ion reactions, which perform differently under high ionic strength; the different reaction space; and the surface of the aerosol and bulk solution. The aerosol water content is scarce compared with the cloud water content, and the low pH values would limit the solubility of SO_2 . The surface-area-to-volume ratio increases with the decreasing of the aerosol diameter, making the chemistry on the surface more important than that in the bulk phase.

Page 8, line 174

The Mn-catalyzed aqueous phase oxidation of S(IV) has been extensively investigated and the critical oxidation process is the reaction between Mn(III) and hydrogen sulfite or its complex with Mn(II). Then SO_3^- radicals can react with dissolved

O₂ to generate SO₅⁻ radicals, which can oxidize Mn(II) to regenerate Mn(III). The bulk manganese catalytic reaction is the ion-ion reaction, in which rate constant decreases dramatically with increasing ionic strength, influenced by the primary kinetic salt effect¹. The enhanced reaction rate for Mn-catalytic reaction on the aerosol surface compared to aqueous reaction is mainly attributed to several differences: the difference between the ion-neutral molecule and ion-ion reactions, which perform differently under high ionic strength; the different reaction space; and the surface of the aerosol and bulk solution. The aerosol water content is scarce compared with cloud water content, and the low pH values limit the solubility of SO₂. The surface-area-to-volume ratio increases with the decreasing of the aerosol diameter, making the chemistry on the surface more important than in the bulk phase. Furthermore, Yan et al.² and Zhang et al.³ found that reaction occurring in microdroplets obtained a higher rate than the same reaction in bulk solution. Hung et al.⁴ proposed sulfate formation in the reaction of sulfurous acid microdroplet and oxygen, without additional oxidants, and the finding of spontaneously H₂O₂ production in the atomized bulk water microdroplets by Lee et al.⁵ may shed light on this phenomenon. These findings demonstrated that there may be a massive gap in the reaction rate between air-liquid interface and bulk solution, and the mechanism of air-liquid interface rate enhancement effect may differ greatly in a distinct reaction system.

The TEM images of Fig 1c are difficult to understand. Please state exactly how they show the Mn-catalytic reaction.

Response: We added a more detailed description to Fig. 1c in our revised manuscript.

Page 5, line 111

Most of the particles containing Mn generated a much larger size (a minimum of 100 nm) during chamber reactions (Supplementary Table 1) compared to the initial produced seed particle size (50 nm). N and S elements observed at the same location indicated the formation of ammonium sulfate.

The evidence for a surface driven reaction seems mainly to be from the Raman single particle analysis that showed increase sulfate mass scales with inverse of particle diameter. Is this true? If so, how does a wet drop sitting on a particle surface, the conditions of the Raman experiment, affect this observation. Eg, how was the particle diameter determined, what is the surface area of a drop sitting on a surface?

Response: We thank the reviewer for raising this question, and we proved surface-driven reactions by two aspects. First, as depicted in Fig. 1a, we observed fast sulfate formation in the presence of Mn, and this high reaction rate could not be explained by aqueous phase model calculation, even without considering inhibition effect by high ionic strength. Chamber experiment results

demonstrated that the proposed reaction is not an aqueous phase reaction. Second, to directly prove the mechanism of heterogenous reaction, we conducted confocal Raman microscope experiments. Based on the phenomenon that the sulfate production rate increased with decreasing droplet diameter, the reaction rates were correlated with the surface-area-to-volume ratio, consistent with the heterogeneous reaction process. In the experiment set-up (Supplementary Fig. 5), droplets of different sizes were sprayed on the PTFE tape substrate placed in the reactor, and were blown upon by an 85% RH carrier gas throughout the experiment time. The droplet diameter was measured by the 50× objective of the Leica DMLM microscope (Fig. 2). Upon your concern, we applied contact angle measurements by an optical tensiometer (Theta Flex, Biolin Scientific) under the same conditions as that of the Raman test (same PTFE tape as substrate, 3 mol L⁻¹ NH₄Cl solution as droplet). As depicted in Supplementary Fig. 6, the contact angles of droplets on PTFE substrate are 112.59° and 112.73°, and the surface area and volume of the droplets could be estimated by the contact angle and measured diameter as a spherical cap:

$$S_{droplet} = \pi dh \quad (\text{Eq.1})$$

$$V_{droplet} = \frac{\pi h^2}{3} \left(\frac{3}{2}d - h \right) \quad (\text{Eq.2})$$

$$h = \frac{d}{2} + \frac{d}{2} \cdot \sin(\theta - 90^\circ) = \frac{d}{2} \cdot (1 - \cos \theta) \quad (\text{Eq.3})$$

$$\frac{S_{droplet}}{V_{droplet}} = -\frac{12}{(\cos \theta + 2)(\cos \theta - 1)} \cdot d^{-1} = 5.37 \cdot d^{-1} \quad (\text{Eq.4})$$

where $S_{droplet}$ is the surface area of the droplet on the PTFE substrate, $V_{droplet}$ is the volume of the droplet, d is the droplet diameter measured by microscope of the Raman system, and h is the height of the spherical cap calculated by the contact angle $\theta = 112.59^\circ$. The surface-area-to-volume ratio could be probed and is only related to droplet diameter. Overall, due to the hydrophobic characteristic of PTFE substrate, the droplets could be exposed to air as much as possible, and the droplet diameter could effectively represent droplet surface area.

We expanded the description of the droplet in the micro-Raman system in the manuscript method part.

Page 23, line 417

Droplets were sitting on the PTFE substrate and had a nearly spherical shape and a contact angle of 112.59° (Supplementary Fig. 6) measured by an optical tensiometer (Theta Flex, Biolin Scientific).

Supplementary Fig. 6. Contact angle of a 3 mol L⁻¹ NH₄Cl droplet on PTFE substrate.

Finally, and of the most importance, if this mechanism is a major route than most of the PM_{2.5} sulfate in ambient particles during winter haze must be internally mixed with Mn²⁺ (or just Mn). The authors seem to have the capability to do single particle analysis, ie Raman or TEM. Or maybe they could collaborate with researchers how have the instrument capable of making this measurement. Alternatively, one could use a cascade impactor to compare the size distribution of sulfate to Mn. I believe size distribution data during these events are available, (published). The authors should check if these data exists and how they support this proposed mechanism, example, maybe one could see if there is sufficient Mn in the size ranges where most the sulfate is found to make this a viable mechanism, based on their chemical model. Since both sulfate and Mn are nonvolatile, sampling issues should not confound the experiment. This is a unique situation, the authors can directly test if their hypothesis is valid (internally sulfate – Mn mixing is a necessary, but not sufficient condition), all they have to do is make the measurement or get the data somehow.

Response: We thank the reviewer's comments on using more field observational evidence to support our results and giving us advice to how to remedy it. We conducted both size distribution of Mn and sulfate measurement and single particle analysis, i.e., TEM. The added evidences are listed as following.

Firstly, to acquire the size distribution of sulfate and Mn concentration, we conducted field observation with cascade impactor in wintertime in Beijing. Different from PM_{2.5} sampling, the cascade impactor may require a longer

sampling time and more polluted condition to reach the limit of detection of corresponding instrument (Ion Chromatography and X-ray Fluorescence in this work) for all stage samples. Therefore, cascade impactor samples were relatively limited, and sampling time might last more than one day. The sampling site was located in northern Beijing (Institute of Atmospheric Physics Chinese Academy of Science), sampling time was Jan. 28th–29th, 2013 (24 h), Feb. 24th–26th, 2014 (48 h), and Dec. 19th–20th, 2016 (24 h). The daily averaged mass concentration of PM_{2.5} during the sampling period were 300, 200, 260, 354, 316, 223, and 376 $\mu\text{g m}^{-3}$, respectively, demonstrating heavy polluted conditions. The concentration of Mn and sulfate shared identical size distribution peak and variation trends in accumulation mode (Supplementary Fig. 11), indicating that sulfate formation process was strongly correlated with Mn.

Secondly, in single particle analysis, we conducted single particle analysis in 2017 winter in Beijing. We found Mn existence in S-rich and Fe-rich particles mixture (Supplementary Fig. 12). Internally mixed sulfate–Mn were not able to be observed because Mn was not detected in S-rich particle due to its trace concentration and the limit of detection of EDX. However, the detection of Fe indicated that metal ions were dissolved in S-rich particle. Given the solubility of Mn is greater than that of Fe in AWC, Mn was likely to exist in S-rich particles. Also, in Dr. Li's published work (Li et al., 2017), in the EDS spectra of particles collected during the research cruise over the Yellow Sea (between mainland China and the Korean Peninsula) in 2013, Mn peaks (5–6 keV) could be observed in the Fe-rich part although they did not add annotation, and Fe was also observed in the S-rich part. This also indicated the Mn commonly exist in sulfate particles. Furthermore, we collected single particle samples on TEM grids in the North China Plain (Shijiazhuang, southwest of Beijing) in Oct. 19th 2020, the TEM and element maps of Fe, Mn, S, O, and N images showed that Fe and Mn are surrounded by S (Supplementary Fig. 13), indicating sulfate formation on the surface of particle containing Fe and Mn. Moreover, Mn appeared to be dispersed in the particles, which proved that Mn and sulfate were internally mixed. These phenomena supplied solid field observational proofs to our proposed mechanism being the dominant sulfate formation pathway in wintertime NCP.

Finally, we conducted water-soluble Mn measurements of sample filters collected in two cities in the Northern China Plain (Baoding and Tianjin) from Jan. 1st to 12th, 2015 in the same period of model simulation (modeling sites and time are shown in Supplementary Fig. 11). The strong correlation between water-soluble Mn and sulfate furtherly demonstrated the viability of our proposed mechanism (Fig. 5).

Furthermore, we conducted water-soluble Mn concentration in the NCP during winter. Compared with total Mn, water-soluble Mn precisely tracked all of the sulfate variation trends in the observation period. (Fig. 5). Since these results were consistent with our prior experiments, they provided solid evidence that dissolved Mn played a key role in sulfate formation. The size distribution of Mn and sulfate content sampled in wintertime Beijing in 2013, 2014, and 2016 showed a similar variation trend in accumulation mode particles during different polluted periods (Supplementary Fig. 11), implying that the formation process of sulfate could be related closely to Mn. Also, we found the presence of Mn in Fe-rich particle of the mixture of Fe-rich and S-rich particle collected in wintertime Beijing and measured by TEM-EDX (Supplementary Fig. 12), although the detection limit of EDX restricted the detection of trace Mn in the S-rich part. Fe detected in the S-rich part could also indicate the existence of dissolved metal in sulfate because the solubility of Mn was greater than Fe in AWC. The TEM and element maps of samples collected in NCP (Shijiazhuang) showed that Fe and Mn were surrounded by S (Supplementary Fig. 13), indicating sulfate formation occurs with the particle containing Fe and Mn. Moreover, Mn appeared to be dispersed in the particle, which proved that Mn and sulfate were internally mixed, indicating that the formation of sulfate could happened with Mn.

Supplementary Fig. 11. Size distributions of sulfate and Mn in aerosol in Beijing. Sampling periods: (a) Jan. 28th–29th, 2013, (b) Feb. 24th–26th, 2014, and (c) Dec. 19th–20th, 2016. The red line represents sulfate content; the blue line represents Mn content.

Supplementary Fig. 12. Morphology and element distribution of a single particle sample collected in urban Beijing, Dec. 30th, 2017. (a) TEM image of sample, (b) EDX spectrum of the S-rich part (red line) and the Fe-rich part (black line). The sampling site was located at the China University of Mining and Technology-Beijing.

Supplementary Fig. 13. TEM image and element maps of a single particle sample collected in Shijiazhang (Hebei Province), Oct. 19th, 2020. (a) refers to the morphology of the sample. (b) refers to the element mapping of the sample. The sample located was the Hebei Normal University.

Fig. 5. Temporal variation of water-soluble Mn, total Mn, and sulfate concentration in PM_{2.5} in the NCP (Baoding and Tianjin) from Jan. 1st to Jan. 12th, 2015. (a) refers to Baoding, and (b) refers to Tianjin. Metal ions and sulfate concentration were measured by inductively coupled plasma mass spectrometry and ion chromatography, respectively.

Reference

1. Herrmann H. Kinetics of aqueous phase reactions relevant for atmospheric chemistry. *Chemical reviews* 103, 4691-4716 (2003).
2. Yan X, Cheng H, Zare RN. Two-phase reactions in microdroplets without the use of phase-transfer catalysts. *Angewandte Chemie* **129**, 3616-3619 (2017).
3. Zhang Y, Apsokardu MJ, Kerecman DE, Achtenhagen M, Johnston MV. Reaction Kinetics of Organic Aerosol Studied by Droplet Assisted Ionization: Enhanced Reactivity in Droplets Relative to Bulk Solution. *Journal of the American Society for Mass Spectrometry*, (2020).
4. Hung HM, Hsu MN, Hoffmann MR. Quantification of SO₂ Oxidation on Interfacial Surfaces of Acidic Micro-Droplets: Implication for Ambient Sulfate Formation. *Environ Sci Technol* **52**, 9079-9086 (2018).
5. Lee JK, *et al.* Spontaneous generation of hydrogen peroxide from aqueous microdroplets. *Proceedings of the National Academy of Sciences* **116**, 19294-19298 (2019).
6. Deguillaume L, Leriche M, Desboeufs K, Mailhot G, George C, Chaumerliac N. Transition metals in atmospheric liquid phases: Sources, reactivity, and sensitive parameters. *Chemical Reviews* 105, 3388-3431 (2005).
7. Fronaeus S, Berglund J, Elding LI. Iron– manganese redox processes and synergism in the mechanism for manganese-catalyzed Autoxidation of hydrogen sulfite. *Inorganic chemistry* 37, 4939-4944 (1998).
8. Berglund J, Elding LI. Manganese-catalysed autoxidation of dissolved sulfur dioxide in the atmospheric aqueous phase. *Atmospheric Environment* 29, 1379-1391 (1995).
9. Li W, *et al.* Air pollution–aerosol interactions produce more bioavailable iron for ocean ecosystems. *Science advances* **3**, e1601749 (2017).

REVIEWERS' COMMENTS

Reviewer #1 (Remarks to the Author):

The authors have done an excellent job in addressing the reviewers comments and revising the manuscript. The manuscript is now ready to be published as is.

Reviewer #2 (Remarks to the Author):

The authors have answered all of my questions seriously via additional experiments. Especially, they compared Fe-catalysis to Mn-catalysis on the basis of new datas. I thus suggest that this manuscript should be published on NC.

Reviewer #3 (Remarks to the Author):

The authors have done a very complete job of addressing the issues raised in the first review. I have only minor comments.

Line 36, change in-filed to in-field

In the discussion on why the reaction occurs at the drop surface versus in the bulk (or throughout) the droplet, what about the issue of dissolved O₂, which is required and also is introduced into the system through the drop surface. Is there a possible gradient in O₂ in the droplet that limits the extent of this chemistry in the droplet bulk. A counter argument to this may be that the concentration of gas phase O₂ is so high that it is not rate limiting anywhere in the drop. This is only a suggestion, I have not investigated this and the authors may wish to ignore this comment.

Overall, a nice paper. Rodney Weber

Point-by-Point Response #2

Please find our detailed point-by-point response to reviewers' comments below. The original comments are in black, our responses are in blue, and the contents added to the revised manuscript are *in italicized blue*.

Reviewer #1 (Remarks to the Author):

The authors have done an excellent job in addressing the reviewers comments and revising the manuscript. The manuscript is now ready to be published as is.

Response: We thank the reviewer for the valuable comments to improve this manuscript.

Reviewer #2 (Remarks to the Author):

The authors have answered all of my questions seriously via additional experiments. Especially, they compared Fe-catalysis to Mn-catalysis on the basis of new datas. I thus suggest that this manuscript should be published on NC.

Response: We thank the reviewer for the valuable comments to improve this manuscript.

Reviewer #3 (Remarks to the Author):

The authors have done a very complete job of addressing the issues raised in the first review. I have only minor comments.

Response: We thank the reviewer for the valuable comments to improve this manuscript. Please find our detailed response below.

Line 36, change in-filed to in-field

Response: Fixed, sorry for the typo.

In the discussion on why the reaction occurs at the drop surface versus in the bulk (or throughout) the droplet, what about the issue of dissolved O₂, which is required and also is introduced into the system through the drop surface. Is there a possible gradient in O₂ in the droplet that limits the extent of this chemistry in the droplet bulk. A counter argument to this may be that the concentration of gas phase O₂ is so high that it is not

rate limiting anywhere in the drop. This is only a suggestion, I have not investigated this and the authors may wish to ignore this comment.

Response: We thank the reviewer for raising this point as to whether the different Mn-catalytic reaction site (surface versus bulk) is related to the issue of dissolved O₂.

In the mechanism for the Mn-catalyzed oxidation of S(IV), O₂ involved in the reaction of oxidizing SO₃⁻ radicals to SO₅⁻ radicals. In the research of aqueous phase oxidation of HSO₃⁻ by O₂, Connick et al. (1995) found that the rate of the reaction is independent of the oxygen concentration. Consistently, in their later work (Connick et al., 1996) investigating Mn-catalyzed oxidation of HSO₃⁻ by O₂ in aqueous phase, they also find linear decrease of O₂ at the steady state of the reaction, demonstrating that the reaction rate is zero-order dependence on oxygen concentration. In the aqueous phase TMI-catalytic rate coefficient measurements, the initial concentration of S(IV) normally ranges from 10⁻⁶ ~ 10⁻⁵ mol L⁻¹. Based on Henry's constant of O₂, the initial dissolved O₂ concentration of 2×10⁻⁴ mol L⁻¹ is in excess of S(IV) concentration, and air or oxygen is persistently introduced into the solution to maintain the dissolved O₂ concentration throughout the experiments. Similarly, in our chamber experiment system, the concentration of S(IV) in droplet bulk phase is estimated as 5×10⁻⁶ ~ 5×10⁻⁵ mol L⁻¹ in the pH range of 3.5 ~ 4.5, the concentration of dissolved O₂ is sufficient both on the surface or in the bulk phase by exposing to zero air. Combined with the conclusion that the rate of reaction is independent of O₂ concentration, therefore, in our perspective, the issue of dissolved O₂ is less possible to be considered as a limiting factor for the low reaction rate of Mn-catalytic aqueous phase reaction.

We added a description in the mechanism of Mn-catalyzed reaction on aerosol surfaces section.

Page9, Line 191

, and the rate of the reaction is independent of the oxygen concentration.

Overall, a nice paper. Rodney Weber

References:

1. Connick RE, Zhang Y-X, Lee S, Adamic R, Chieng P. Kinetics and mechanism of the oxidation of HSO₃⁻ by O₂. 1. the uncatalyzed reaction. *Inorganic Chemistry* **34**, 4543-4553 (1995).
2. Connick RE, Zhang Y-X. Kinetics and mechanism of the oxidation of HSO₃⁻ by

O₂. 2. the manganese (II)-catalyzed reaction. *Inorganic Chemistry* **35**, 4613-4621 (1996).